# Inv-Entropy: A Fully Probabilistic Framework for Uncertainty Quantification in Language Models

**Haoyi Song**
University of Michigan
haoyiso@umich.edu

**Ruihan Ji**
University of Minnesota
ji000234@umn.edu

**Naichen Shi**
Northwestern University
naichen.shi@northwestern.edu

**Fan Lai**
University of Illinois Urbana-Champaign
fanlai@illinois.edu

**Raed Al Kontar**[*]
University of Michigan
alkontar@umich.edu

## Abstract

Large language models (LLMs) have transformed natural language processing, but their reliable deployment requires effective uncertainty quantification (UQ). Existing UQ methods are often heuristic and lack a probabilistic interpretation. This paper begins by providing a theoretical justification for the role of perturbations in UQ for LLMs. We then introduce a dual random walk perspective, modeling input–output pairs as two Markov chains with transition probabilities defined by semantic similarity. Building on this, we propose a fully probabilistic framework based on an inverse model, which quantifies uncertainty by evaluating the diversity of the input space conditioned on a given output through systematic perturbations. Within this framework, we define a new uncertainty measure, Inv-Entropy. A key strength of our framework is its flexibility: it supports various definitions of uncertainty measures, embeddings, perturbation strategies, and similarity metrics. We also propose GAAP, a perturbation algorithm based on genetic algorithms, which enhances the diversity of sampled inputs. In addition, we introduce a new evaluation metric, Temperature Sensitivity of Uncertainty (TSU), which directly assesses uncertainty without relying on correctness as a proxy. Extensive experiments demonstrate that Inv-Entropy outperforms existing semantic UQ methods. The code to reproduce the results can be found at https://github.com/UMDataScienceLab/Uncertainty-Quantification-for-LLMs.

## 1 Introduction

Large language models (LLMs) have demonstrated remarkable success in various natural language processing tasks, such as text generation, question answering, and summarization [Brown et al., 2020, Chowdhery et al., 2023, Touvron et al., 2023]. These models have pushed the boundaries of what is achievable in language understanding and generation [Chung et al., 2024, OpenAI, 2023]. However, despite their impressive capabilities, a significant challenge remains: LLMs tend to hallucinate, or generate confidently wrong predictions [Maynez et al., 2020, Zhang et al., 2024b]. This is a serious concern in applications where reliability is paramount, such as healthcare, autonomous systems, and legal domains, where incorrect outputs can have dire consequences [Ji et al., 2023]. Addressing these challenges is essential for ensuring the reliable and responsible deployment of LLMs at scale, ultimately unlocking their full potential in real-world applications.

---

[*]Corresponding author.

39th Conference on Neural Information Processing Systems (NeurIPS 2025).

A central step in addressing these limitations is developing effective measures for UQ, enabling LLMs to acknowledge their confidence in a generated output. Existing UQ approaches rely predominantly on heuristic consistency checks. Most commonly, they use the model's own generation likelihood or perplexity as a proxy for confidence, or they measure dispersion across multiple sampled continuations (e.g. via n-gram overlap or embedding-space variance) [Mudumbai and Bell, 2024]. Such likelihood-based and sampling-based measures, however, lack a grounded probabilistic justification. Also, token-level probabilities are known to dramatically under-estimate uncertainty as models can be "confidently wrong" [Jiang et al., 2021], and they are often inaccessible in black-box LLMs.

To probe deeper LLM brittleness, recent work has turned to input-perturbation UQ methods. Variant prompts are created through paraphrasing, adversarial token insertion, or temperature shifts. Output sensitivity is then quantified as an uncertainty signal [Gao et al., 2024, Tuna et al., 2022, Seeböck et al., 2019]. For instance, Gao et al. [2024] randomly perturb both prompt wording and sampling temperature, then aggregate variation across outputs to flag unstable predictions. Tuna et al. [2022] applies adversarial paraphrases to uncover "blind spots" where small semantic-preserving edits trigger large output changes, while Seeböck et al. [2019] uses systematic character-level and word-level corruptions to map regions of high model vulnerability.

Despite these advances, an *ab initio* probabilistic framework has not been established, as existing methods mainly rely on heuristic score functions for UQ. To address this, we introduce Inv-Entropy, a fully probabilistic framework built on random-walk theory that offers a new perspective: it learns the statistical connections between LLM inputs and outputs. This is accomplished through structured perturbations, which simultaneously capture input variability and the influence of different inputs on model predictions. Our contributions are summarized as follows:

1. We present the first work in UQ for LLMs that adopts a **fully probabilistic framework**, grounded in random walk theory. This framework is highly flexible, and its probabilistic nature allows for the use of various UQ measures. It is also intrinsically capable of handling black-box models, as it does not rely on token probabilities.

2. We introduce an **inverse perspective** that quantifies input diversity given an output, inspired by "Asymmetry in Semantic Variability" (defined later). Extensive simulations highlight its advantages, especially for short-form answers where traditional methods often struggle.

3. Theoretically, our work provides a **principled foundation** for using perturbation-based methods for UQ, justifying the recent momentum behind such approaches.

4. We propose GAAP, a novel perturbation algorithm that enhances **input sampling diversity**. Empirical results show that GAAP significantly improves perturbation-based UQ.

5. Finally, we introduce a new evaluation metric, **Temperature Sensitivity of Uncertainty (TSU)** capable of evaluating uncertainty *without relying on correctness as a proxy*. This enables evaluation of UQ on any dataset, even when labels are unavailable.

For the remainder of the paper, we use $\mathbb{E}[\cdot]$ to denote expectation and $\mathbb{V}[\cdot]$ to denote variance. We note that a more detailed related work section can be found in Appendix A.

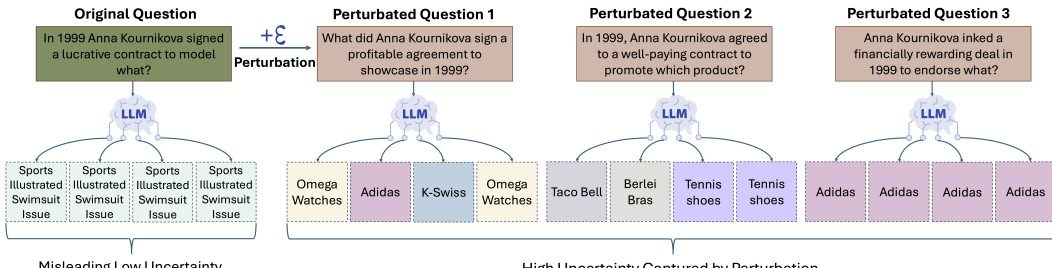

Figure 1: Toy example highlighting the importance of perturbations. The original question is from TriviaQA [Joshi et al., 2017], and the correct answer is "bras." The responses are generated by ChatGPT-3.5-Turbo. Input perturbations reveal hidden variability that multiple sampling (i.e., replications) alone fails to capture, as replication alone can be confidently wrong.

## 2 Perturb-then-Quantify

### 2.1 Why perturb the input?

We begin with a simple illustrative example in Fig. 1 to highlight the importance of input perturbations. In this example, when the input question is simply replicated (as is common in much of the existing literature [Lin et al., 2024]) all generated responses are identical. This misleadingly suggests low uncertainty even though all answers are incorrect, thus providing false confidence. Replication alone is therefore insufficient to capture the underlying variability. In contrast, applying input perturbations yields a diverse set of responses across semantically equivalent prompts, enabling a more faithful characterization of uncertainty. The importance of such perturbations is not only evident from this example but also supported by simplified theoretical argument in what follows.

To build theoretical insight into the importance of perturbations, we leverage a key property specific to language: semantic equivalence. An *ideal* LLM should respond consistently to semantically equivalent inputs. We use this property, in a simplified setting, to construct input perturbations within equivalence classes, which allows us to expose inconsistencies in model behavior and formally reason about uncertainty. We start from a proof-of-concept example where the target function $f^\star$ is a single output function $f^\star : \mathbb{R}^d \to \mathbb{R}$. We lack access to $f^\star$, but have access to an approximation $\hat{f} : \mathbb{R}^d \to \mathbb{R}$, such as a pre-trained model. At first sight, it seems hopeless to quantify the alignment between $\hat{f}$ and $f^\star$ without knowing $f^\star$. However, the semantic equivalence class relative to an input $x_0$ provides a set $\mathcal{I}(x_0) \subseteq \mathbb{R}^d$, such that $f^\star(x') = f^\star(x_0), \forall x' \in \mathcal{I}(x_0)$. The equivalence class provides valuable information for UQ as shown in the subsequent argument and Figure 2.

The shape of $\mathcal{I}(x_0)$ could be complex for general $f^\star$. However, in a small neighborhood of $x_0$ where $\nabla f^\star(x_0) \neq 0$, $\mathcal{I}(x_0)$ should be locally close to the tangent space of $f^\star$. More specifically, we define a tangent invariance set as $\mathcal{I}_{\text{tangent}}(x_0) = \{x_0 + (I - \eta_{\nabla f^\star}) z; z \in \mathbb{R}^d\}$, where $\eta_{\nabla f^\star} = \frac{\nabla f^\star(x_0) \nabla f^\star(x_0)^\top}{\|\nabla f^\star(x_0)\|^2}$. Intuitively, $I - \eta_{\nabla f^\star}$ acts as the orthogonal projector onto the subspace orthogonal to $\nabla f^\star(x_0)$. Taylor expansion shows $\mathcal{I}_{\text{tangent}}(x_0) \approx \mathcal{I}(x_0)$ when restricted to the small neighborhood of $x_0$. To generate algorithmic insight, we further define a probability density on $\mathcal{I}_{\text{tangent}}(x_0)$: we use $P(x'; x_0, \sigma)$ to denote the probability density function of $x' = x_0 + (I - \eta_{\nabla f^\star}) z$ where $z$ is sampled from a $d$-dimensional isotropic Gaussian distribution $\mathcal{N}(0, \sigma^2)$. It is easy to verify that $P(\cdot; x_0, \sigma)$ is a Gaussian distribution supported on $\mathcal{I}_{\text{tangent}}(x_0)$ whose mean is $x_0$.

The following Lemma shows that we can estimate the angle between $\nabla \hat{f}(x)$ and $\nabla f^\star(x)$ by examining the variance of $\hat{f}(x')$, where $x'$ is sampled from $P(x'; x, \sigma)$.

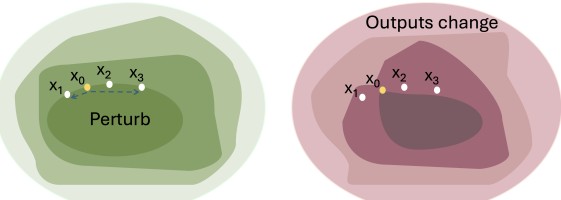

Figure 2: **Left**: Conceptual illustration of level sets of the ground truth $f^\star$. We perturb the input $x_0$ to $x_1, x_2, \ldots$ along a schematic isocontour such that $f^\star(x_0) = f^\star(x_1) = \cdots$. **Right**: Conceptual illustration of level sets of the model $\hat{f}$. The deviations of $\hat{f}(x_i)$ for $i \geq 1$ from $\hat{f}(x_0)$ reflect the model's uncertainty around $x_0$.

**Lemma 2.1** *Assume (1) $\hat{f}$ is twice differentiable, and both $\|\nabla \hat{f}(x)\|$ and $\|\nabla^2 \hat{f}(x)\|_{op}$ are bounded for all $x \in \mathbb{R}^d$, and (2) $\nabla \hat{f}(x_0) \neq 0$ and $\nabla f^\star(x_0) \neq 0$. Then, for sufficiently small $\sigma$, we have*

$$\frac{1}{\sigma^2} \mathbb{V}_{x' \sim \mathcal{P}(\cdot; x_0, \sigma^2)}[\hat{f}(x')] = \|\nabla \hat{f}(x_0)\|^2 \sin^2 \theta \left(\nabla \hat{f}(x_0), \nabla f^\star(x_0)\right) + \mathcal{O}(\sigma), \quad (1)$$

*where $\theta(v_1, v_2) = \arccos\left(\frac{v_1^\top v_2}{\|v_1\| \|v_2\|}\right)$ denotes the angle between two vectors $v_1$ and $v_2$.*

Equation (1) shows that the variance of function values of $\hat{f}$ on the invariance set is indicative of the alignment between $\nabla \hat{f}$ and $\nabla f^\star$. When the variance is larger, $\hat{f}$ respects the invariance of $f^\star$ less,

which in turn implies larger misalignment between $\nabla \hat{f}$ and $\nabla f^\star$ and larger uncertainty. The full proof of Lemma 2.1 is relegated to the Appendix B.

Despite its simplicity, Lemma 2.1 demonstrates two important elements of perturbation-based UQ: input perturbation and output variance evaluation. In what follows, we introduce concrete designs that implement these elements in the LLM context to effectively characterize uncertainty.

## 2.2 A probabilistic framework via dual random walks

The variance estimate in Lemma 2.1 presents useful insights yet is oversimplified to realistically model LLMs, whose outputs are token sequences rather than a single scalar. Also, it is difficult to sample exactly from a semantic equivalence class. To tackle these challenges, we introduce a probabilistic approach inspired by Markov chains.

We use $S_{x_0}$ to denote a semantic input to a LLM $f$, and $S_{y_0} \leftarrow f(S_{x_0})$ to denote one of its possible corresponding outputs. Because $f$ is typically stochastic, the output $S_{y_0}$ is a random variable. Next, consider a semantic embedding function $\psi$ that maps both inputs and outputs into a continuous space $\mathcal{X}$, such that $(x_0, y_0) = (\psi(S_{x_0}), \psi(S_{y_0}))$. We also assume a perturbation algorithm $\mathcal{P}\mathrm{er}(S_{x_0})$ that produces a finite set of perturbed inputs:

$$\mathcal{P}\mathrm{er}(S_{x_0}) = \{S_{x_0}, S_{x_1}, \ldots, S_{x_n}\},$$

whose detailed implementation will be described in Sec. 2.6.

Applying the embedding function $\psi(S_{x_i})$, we obtain the embeddings:

$$X_n = \{x_0, x_1, \ldots, x_n\}, \text{ where } x_i = \psi(S_{x_i}).$$

One corresponding output embedding set is hence given as

$$Y_n = \{y_0, y_1, \ldots, y_n\}, \text{ where } y_i = \psi(S_{y_i}) = \psi(f(S_{x_i})).$$

Samples in $X_n$ and $Y_n$ are one-to-one correspondent. We can thus estimate the structural similarity of the samples in the two sets. In the following, we propose a probabilistic approach to characterize the structural similarity through the lens of a random walk.

We define two Markov chains, $\mathcal{M}_x$ and $\mathcal{M}_y$, over the same state set $\mathcal{S} = \{0, 1, \ldots, n\}$, where each state $i$ corresponds to a perturbed instance $S_{x_i}$. The chains differ in their transition dynamics: $\mathcal{M}_x$ is built from similarities among input embeddings $X_n$, and $\mathcal{M}_y$ from similarities among output embeddings $Y_n$. Let $a_{\mathrm{Similarity}}(x, x') : \mathcal{X} \times \mathcal{X} \to \mathbb{R}_0^+$ denote a non-negative similarity function that measures the closeness between embeddings in $\mathcal{X}$. Using this function, we define the transition matrices $\mathrm{P}_x, \mathrm{P}_y \in \mathbb{R}^{(n+1)\times(n+1)}$ for $\mathcal{M}_x$ and $\mathcal{M}_y$ elementwise as:

$$\mathrm{P}_x[i, j] \triangleq \frac{a_{\mathrm{Similarity}}(x_i, x_j)}{\sum_{k=0}^n a_{\mathrm{Similarity}}(x_i, x_k)}, \quad \mathrm{P}_y[i, j] \triangleq \frac{a_{\mathrm{Similarity}}(y_i, y_j)}{\sum_{k=0}^n a_{\mathrm{Similarity}}(y_i, y_k)}. \tag{2}$$

The two transition probability matrices characterize two random walks in the space of $n + 1$ pairs $\{x_i, y_i\}_{i=0}^n$, where the transition probability from $i$ to $j$ is higher if their input or output semantic features are closer. Notice that a variety of well-established methods have been proposed for $a_{\mathrm{Similarity}}$, some of which are deployed in our numerical studies (see Sec. 3).

At its core, (2) defines stochastic dynamics on the set $\mathcal{S}$, capturing the similarity structures in both the input and output spaces. These dual dynamics uncover meaningful semantic patterns that can be leveraged for UQ. There are various ways to characterize uncertainty by examining the alignment between the two induced graphs [Vishwanathan et al., 2010]. In what follows, we construct a framework tailored for discrete input and output spaces to rigorously define uncertainty.

## 2.3 Constructing the distributions

We use $X$ and $Y$ to denote discrete random variables whose supports are $X_n$ and $Y_n$, respectively. There are many possible ways to define possible distributions of $X$ and $Y$. In the following, we will introduce one design of $P(Y)$ and $P(X|Y)$ based on (2). For notational simplicity, we denote the uniform distribution over all states $\mathcal{S}$ by $\pi^{\mathrm{Uniform}}$, which is given by $\pi^{\mathrm{Uniform}} = \begin{bmatrix} \frac{1}{n+1} & \frac{1}{n+1} & \cdots & \frac{1}{n+1} \end{bmatrix}$.

**The marginal distribution**   The random variable $Y$ corresponds to the LLM's response to a question that is perturbed from the original question $x_0$. We define the marginal distribution of $Y$ as:

$$P(Y = y_j) \triangleq (\pi^{\text{Uniform}} \mathrm{P}_y)[j] = \frac{1}{n+1} \sum_{i=0}^{n} \frac{a_{\text{Similarity}}(y_i, y_j)}{\sum_k a_{\text{Similarity}}(y_i, y_k)}, \tag{3}$$

where notation $[j]$ denotes the $j$-th element of a vector.

The distribution (3) has an intuitive interpretation: we randomly sample a point uniformly from the state space $\mathcal{S}$, then randomly transit the sample with $\mathrm{P}_y$ for one step. After the transit step, nodes whose corresponding output samples are surrounded by many similar outputs are assigned a higher probability, while isolated nodes with fewer similar neighbors are assigned a lower probability. Therefore, the mass is concentrated on regions of high semantic density in the output space.

**The conditional distribution**   Now, we introduce the conditional probability $P(X = x_i \mid Y = y_j)$

$$P(X = x_i | Y = y_j) = (\mathrm{P}_y \mathrm{P}_x)[j, i] = \sum_k \frac{a_{\text{Similarity}}(x_i, x_k)}{\sum_m a_{\text{Similarity}}(x_m, x_k)} \frac{a_{\text{Similarity}}(y_j, y_k)}{\sum_l a_{\text{Similarity}}(y_j, y_l)}, \tag{4}$$

where notation $[j, i]$ denotes the $(j, i)$-th entry of the matrix $\mathrm{P}_y \mathrm{P}_x \in \mathbb{R}^{(n+1) \times (n+1)}$.

Essentially, (4) defines the conditional probability through composite transition dynamics on the state space $\mathcal{S}$. We design a two-stage random walk: first, states transition under $\mathrm{P}_y$, capturing output similarities; then, they transition under $\mathrm{P}_x$, capturing input similarities. Since both $\mathrm{P}_x$ and $\mathrm{P}_y$ operate on the same state space $\mathcal{S}$ linked by the LLM, (4) establishes a probabilistic bridge connecting similarity structures in the output and input spaces (see Fig. 3).

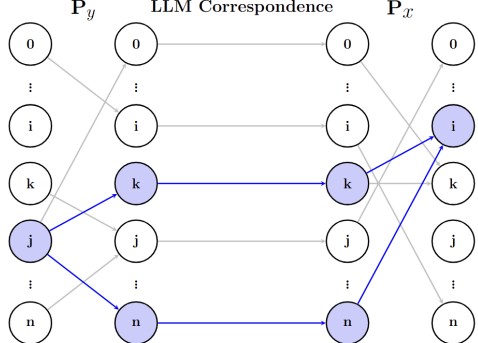

We explicitly model the conditional distribution of inputs $X$ rather than outputs $Y$ for two main reasons. First, the perturbed samples generated by $\mathcal{P}\mathrm{er}(S_{x_0})$ may differ semantically from the original input. Reweighting these samples using both input- and output-space similarities ensures that semantically consistent perturbations are emphasized while spurious ones are down-weighted. Second, LLMs exhibit an inherent *semantic asymmetry* between inputs and outputs: many distinct prompts can lead to similar responses, whereas small input changes typically cause only small to modest output variations. Modeling $P(X \mid Y)$ therefore provides a more stable and informative view of uncertainty by capturing the diversity of possible inputs that could relate to a given output.

Figure 3: Random-walk transitions underlying $P(X \mid Y) = \mathrm{P}_y \mathrm{P}_x$. Highlighted blue paths show two representative transitions (one through $k$ and one through $n$), each following $y_j \xrightarrow{\mathrm{P}_y} y_k \xrightarrow{\text{LLM}} x_k \xrightarrow{\mathrm{P}_x} x_i$.

Our goal is to use the information encoded in $Y$ to guide the conditional distribution of $X$. This coupling between input and output similarities allows the model to assign higher probability to inputs supported by multiple semantically consistent input-output pairs, yielding a more faithful representation of uncertainty. Based on the defined $P(Y)$ and $P(X \mid Y)$, we can then derive $P(X) \triangleq \pi^{\text{Uniform}} \mathrm{P}_y \mathrm{P}_x$ and $P(Y \mid X)$ via Bayes' theorem. Such formulation provides a flexible foundation for defining diverse uncertainty measures, including divergence-based metrics (e.g., KL or Wasserstein distances), entropy-based quantities, and other probabilistic constructs. In the next section, we introduce an entropy-based metric as a concrete example.

## 2.4   Inv-Entropy via bootstrapping and Monte Carlo

We next introduce how to leverage the probabilistic framework described above to define our UQ measure, denoted as Inverse-Entropy (Inv-Entropy).

---

**Algorithm 1** Inv-Entropy overall framework

---
1: **Input:** $(S_{x_0}, S_{y_0})$, $f$, $\psi$, $\mathcal{P}er$, and $a_{\text{Similarity}}$
2: **Perturb**: Use $\mathcal{P}er(S_{x_0})$ and $\psi$ to obtain $\tilde{X}_n$, Compute $P_x$ using (2)
3: **LLM Generation**: Input each question in $\mathcal{P}er(S_{x_0})$, $r$ times into $f$ and obtain $\{\mathcal{R}_i\}_{i=0}^n$.
4: **for** $b = 1$ to $B$ **do**
5:     Sample $Y_n^{(b)} = \{y_0^{(b)}, \ldots, y_n^{(b)}\}$
6:     Compute $P_y^{(b)}$ using (2)
7:     Compute $P^{(b)}(x_i|y_i) = (P_y^{(b)} \cdot P_x)[i, i]$ (from (4)) for $\forall\, i$
8:     Compute $H(X_n \mid Y_n^{(b)})$ using (5)
9: **end for**
10: Compute $\widehat{H}(X|Y)$ using (6)

---

**Inv-Entropy**    A natural measure of uncertainty is the conditional sample entropy $H(X_n \mid Y_n)$, which connects the similarity of the input set $X_n$ to the similarity of the corresponding output set $Y_n$:

$$H(X_n \mid Y_n) \triangleq -\,\text{trace}\big(P_y P_x \odot \log(P_y P_x)\big) = -\sum_{i=0}^n P(x_i \mid y_i) \log P(x_i \mid y_i), \qquad (5)$$

where $\odot$ denotes the Hadamard product. This quantity captures the degree of alignment between $P_y$ and $P_x$. High entropy indicates that semantically similar inputs yield divergent outputs, or that semantically distinct outputs correspond to similar inputs, both revealing uncertainty in the model's behavior. Note that Inv-Entropy represents an *unnormalized* entropy measure, designed to capture not only the dispersion of $P(x_i \mid y_i)$ but also its magnitude, thereby jointly modeling these two sources of uncertainty.

That said, it is important to realize that $X_n$ does not map to a unique $Y_n$ due to the inherent stochasticity of $f$ as LLMs can produce different outputs for the same input. Thus, beyond perturbations that capture epistemic uncertainty, replications can help account for aleatoric uncertainty arising from sampling randomness. To this end, each perturbed input $S_{x_i} \in \mathcal{P}er(S_{x_0})$ can be queried $r$ times, yielding a replicated output set $\mathcal{R}_i = \{y_{i,1}, \ldots, y_{i,r}\}$ for each question $x_i$, where $y_{i,\cdot} = \psi(f(S_{x_i}))$.

**Bootstrapping & Monte Carlo**    With this setup, we are able to generate $B$ bootstrap samples $Y_n^{(b)} = \{y_0^{(b)}, \ldots, y_n^{(b)}\}$ for $b \in \{1, \ldots, B\}$, each corresponding to $X_n$, where $y_i^{(b)} \sim \mathcal{R}_i$ is drawn with replacement for $i \in \{0, \ldots, n\}$, yielding a transition matrix $P_y^{(b)}$. Each bootstrapped output embedding set $Y_n^{(b)}$, together with the input embeddings $X_n$, defines a probabilistic instance of our uncertainty measure, Inv-Entropy. The final UQ estimate is obtained by averaging Inv-Entropy over the $B$ bootstrap replicates:

$$\widehat{H}(X \mid Y) \triangleq \frac{1}{B} \sum_{b=1}^B H(X_n \mid Y_n^{(b)}). \qquad (6)$$

Our overall framework in given in Algorithm 1. It is important to emphasize that a key strength of our framework lies in its flexibility: it accommodates arbitrary choices of embedding function $\psi$, perturbation strategy $\mathcal{P}er$, and similarity metric $a_{\text{Similarity}}$, allowing it to adapt to different tasks and model architectures. Moreover, $\widehat{H}(X \mid Y)$ represents just one possible uncertainty measure; many others can be defined within our probabilistic framework. For instance, our numerical studies evaluate alternatives such as the Wasserstein distance between the marginal distributions $P(X)$ and $P(Y)$.

## 2.5   Insights into Inv-Entropy

To provide some insights into our UQ measure, we introduce two parameters: $\epsilon_x \in (0, 1]$, which controls the input perturbation level, and $\epsilon_y \in [0, 1]$, which controls the output dispersion level (defined via $a_{\text{Similarity}}$ below). The corresponding transition matrices are then readily derived as:

$$a_{\text{Similarity}}(z_i, z_j) = \begin{cases} 1, & i = j, \\ 1 - \epsilon_z, & i \neq j, \end{cases} \rightarrow P_z(\epsilon_z)[i, j] = \begin{cases} \dfrac{1}{(n+1) - n\epsilon_z}, & i = j, \\ \dfrac{1 - \epsilon_z}{(n+1) - n\epsilon_z}, & i \neq j, \end{cases}$$

where $z \in \{x, y\}$. With this, the joint conditional probability can be written as: $P(x_i \mid y_i; \epsilon_x, \epsilon_y) = \mathbf{P}_y(\epsilon_y)\mathbf{P}_x(\epsilon_x)[i, i] = \frac{1+n-n\epsilon_x-n\epsilon_y+n\epsilon_x\epsilon_y}{(n+1-n\epsilon_x)(n+1-n\epsilon_y)}$. We can then show that for the same input perturbation level $\epsilon_x$, a larger output dispersion $\epsilon_y$ corresponds to higher uncertainty, consistent with the definition of Inv-Entropy. To see this, notice that the derivative of $P(x_i \mid y_i; \epsilon_x, \epsilon_y)$ with respect to $\epsilon_y$ is $\frac{\partial P(x_i \mid y_i; \epsilon_x, \epsilon_y)}{\partial \epsilon_y} = \frac{n\epsilon_x}{(n+1-n\epsilon_x)(n+1-n\epsilon_y)^2} > 0$. Now, if we assume all $P(x_i \mid y_i; \epsilon_x, \epsilon_y)$ are smaller than $\frac{1}{e}$ (a condition generally satisfied for large $n$), then the Inv-Entropy function $H$ in (5) becomes also increasing in $P(x_i \mid y_i; \epsilon_x, \epsilon_y)$ and hence in $\epsilon_y$.

## 2.6 GAAP

In this section, we present a genetic algorithm-based adversarial perturbation (GAAP) that progressively modifies the semantic input $S_{x_0}$ to generate controlled perturbations, $\mathcal{P}er(S_{x_0})$. Below we highlight our overarching framework, while algorithmic details are relegated to Appendix C. As shown in Fig. 4, the process consists of an initialization step and multiple iterative procedures. In the initialization step, we construct a population $\text{Pop}_0(S_{x_0})$ consisting of perturbed versions of $S_{x_0}$. More specifically, each text in $\text{Pop}_0(S_{x_0})$ is derived from $S_{x_0}$ by replacing one keyword with a synonym, hypernym, hyponym from WordNet [Miller, 1995], or a deletion.

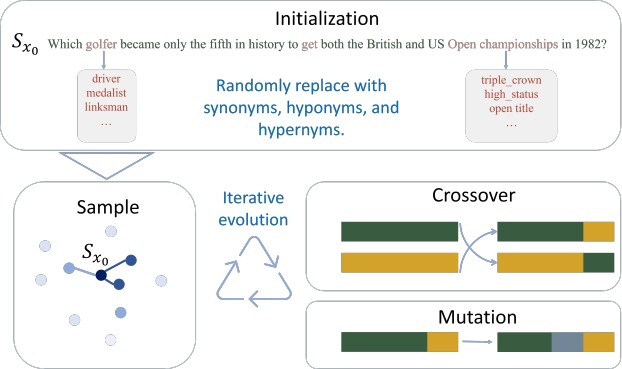

Figure 4: Illustration of GAAP on a TriviaQA [Joshi et al., 2017] question.

With an initial $\text{Pop}_0(S_{x_0})$, GAAP updates the $\text{Pop}_t(S_{x_0})$ through subsequent steps of crossovers and mutations. In the crossover step, we first select a random subset of $\text{Pop}_t(S_{x_0})$ based on $a_{\text{Similarity}(x_0, \cdot)}$, such that samples are chosen with higher probability if they are closer to $x_0$. Next, we randomly segment each selected sentence. These sentence segments are then randomly concatenated to generate new sentences, as illustrated in Crossover of Fig. 4.

In the mutation step, we perturb the recombined sentences with further key word substitutions and deletions, which introduce additional variations to the cross-pollinated texts. We construct the next generation population $\text{Pop}_{t+1}(S_{x_0})$ as the union of the selected, crossovered, and mutated texts.

GAAP proceeds by iteratively updating $\text{Pop}_t(S_{x_0})$ with crossovers and mutations for $T$ iterations or until all texts in $\text{Pop}_t(S_{x_0})$ have similarity to $S_{x_0}$ smaller than a predefined constant $\delta$. Finally, we construct $\mathcal{P}er(S_{x_0})$ by sampling from populations at different generations $\{\text{Pop}_t(S_{x_0})\}_{t=0, \tau, 2\tau, \cdots}$, where $\tau \in \mathbb{Z}$ is a fixed gap. Since texts in $\text{Pop}_t(S_{x_0})$ tend to deviate from $S_{x_0}$ further with larger $t$, such construction of $\mathcal{P}er(S_{x_0})$ ensures diverse representation of perturbed texts with different levels of similarity to $S_{x_0}$.

## 3 Experiments

**Models and tasks** We conducted experiments using two language models: GPT-3.5-Turbo, a black-box model accessed via API, and LLaMA-3.1-8B-Instruct, a grey-box model. We evaluated our framework on datasets spanning three categories: question answering (TriviaQA [Joshi et al., 2017], SciQ [Welbl et al., 2017], Natural Questions [Kwiatkowski et al., 2019] (NQ, long-answer questions with details in Appendix D.3)), multiple choice (MMLU [Hendrycks et al., 2020]), and mathematical reasoning (GSM8K [Cobbe et al., 2021]).

Table 1: Comparison of AUROC, PRR, and Brier scores across all the 5 datasets. We use GPT-3.5-Turbo with ChatGPT-based paraphrasing, and DeBERTa-v2-xlarge-MNLI embedding function. **Bold** and underline denote the best and second-best performers, respectively.

| Metric | Method | Datasets | | | | |
|--------|--------|----------|------|------|------|-------|
| | | TriviaQA | SciQ | NQ | MMLU | GSM8K |
| AUROC (↑) | Semantic Entropy | $0.579 \pm 0.044$ | $0.679 \pm 0.045$ | $0.521 \pm 0.034$ | $0.518 \pm 0.048$ | $0.589 \pm 0.052$ |
| | Kernel Entropy | $0.687 \pm 0.062$ | $0.685 \pm 0.063$ | $0.556 \pm 0.055$ | $0.653 \pm 0.059$ | $0.560 \pm 0.060$ |
| | VU | $0.695 \pm 0.060$ | $0.480 \pm 0.060$ | $0.533 \pm 0.056$ | $0.523 \pm 0.054$ | $0.557 \pm 0.057$ |
| | P(True) | $0.604 \pm 0.050$ | $0.522 \pm 0.026$ | $0.519 \pm 0.020$ | $0.474 \pm 0.027$ | $0.571 \pm 0.056$ |
| | LexSim | $0.649 \pm 0.055$ | $0.681 \pm 0.046$ | $0.518 \pm 0.055$ | $0.643 \pm 0.054$ | $0.598 \pm 0.060$ |
| | DegMat | $0.734 \pm 0.056$ | $0.672 \pm 0.059$ | $0.551 \pm 0.052$ | $0.608 \pm 0.058$ | $0.678$ $\pm 0.059$ |
| | LUQ | $0.637 \pm 0.067$ | $0.726$ $\pm 0.048$ | $0.627 \pm 0.055$ | $0.648 \pm 0.057$ | $0.662 \pm 0.064$ |
| | KLE | $0.333 \pm 0.054$ | $0.341 \pm 0.056$ | $0.410 \pm 0.060$ | $0.360 \pm 0.064$ | $0.338 \pm 0.061$ |
| | Inv-Entropy | **0.788** $\pm 0.054$ | **0.740** $\pm 0.050$ | **0.661** $\pm 0.052$ | **0.780** $\pm 0.041$ | **0.695** $\pm 0.051$ |
| | NI-Entropy | 0.786 $\pm 0.057$ | $0.681 \pm 0.056$ | 0.637 $\pm 0.053$ | 0.710 $\pm 0.052$ | $0.650 \pm 0.069$ |
| | NR-Inv-Entropy | $0.743 \pm 0.061$ | $0.720 \pm 0.049$ | $0.627 \pm 0.054$ | $0.604 \pm 0.059$ | $0.677 \pm 0.064$ |
| | WD-px-py | $0.518 \pm 0.060$ | $0.303 \pm 0.060$ | $0.558 \pm 0.055$ | $0.573 \pm 0.061$ | $0.605 \pm 0.069$ |
| | MAX-py-x | $0.723 \pm 0.054$ | $0.674 \pm 0.054$ | $0.547 \pm 0.051$ | $0.585 \pm 0.059$ | $0.618 \pm 0.059$ |
| PRR (↑) | Semantic Entropy | $0.517 \pm 0.060$ | $0.763 \pm 0.044$ | $0.505 \pm 0.049$ | $0.690 \pm 0.058$ | $0.335 \pm 0.056$ |
| | Kernel Entropy | $0.794 \pm 0.052$ | $0.812 \pm 0.039$ | $0.573 \pm 0.068$ | $0.768 \pm 0.057$ | $0.333 \pm 0.054$ |
| | VU | $0.723 \pm 0.053$ | $0.677 \pm 0.053$ | $0.537 \pm 0.053$ | $0.654 \pm 0.055$ | $0.328 \pm 0.057$ |
| | P(True) | $0.797 \pm 0.042$ | $0.679 \pm 0.050$ | $0.502 \pm 0.050$ | $0.671 \pm 0.041$ | $0.303 \pm 0.056$ |
| | LexSim | $0.810 \pm 0.045$ | $0.770 \pm 0.051$ | $0.563 \pm 0.064$ | $0.767 \pm 0.053$ | $0.356 \pm 0.076$ |
| | DegMat | $0.882 \pm 0.041$ | $0.802 \pm 0.046$ | $0.549 \pm 0.069$ | $0.771 \pm 0.058$ | $0.462 \pm 0.091$ |
| | LUQ | $0.854 \pm 0.043$ | $0.840 \pm 0.045$ | 0.595 $\pm 0.066$ | $0.787 \pm 0.052$ | 0.504 $\pm 0.094$ |
| | KLE | $0.704 \pm 0.048$ | $0.592 \pm 0.059$ | $0.449 \pm 0.059$ | $0.612 \pm 0.061$ | $0.224 \pm 0.043$ |
| | Inv-Entropy | **0.885** $\pm 0.044$ | **0.853** $\pm 0.042$ | **0.614** $\pm 0.067$ | **0.898** $\pm 0.030$ | **0.521** $\pm 0.094$ |
| | NI-Entropy | 0.883 $\pm 0.043$ | $0.781 \pm 0.053$ | $0.592 \pm 0.064$ | 0.823 $\pm 0.055$ | $0.501 \pm 0.098$ |
| | NR-Inv-Entropy | $0.840 \pm 0.054$ | 0.844 $\pm 0.045$ | $0.576 \pm 0.069$ | $0.743 \pm 0.064$ | $0.518 \pm 0.087$ |
| | WD-px-py | $0.763 \pm 0.051$ | $0.587 \pm 0.056$ | $0.586 \pm 0.065$ | $0.777 \pm 0.054$ | $0.420 \pm 0.085$ |
| | MAX-py-x | $0.875 \pm 0.038$ | $0.821 \pm 0.048$ | $0.536 \pm 0.066$ | $0.749 \pm 0.062$ | $0.413 \pm 0.081$ |
| Brier (↓) | Semantic Entropy | $0.166 \pm 0.023$ | $0.173 \pm 0.020$ | $0.242 \pm 0.006$ | $0.208 \pm 0.020$ | $0.188 \pm 0.018$ |
| | Kernel Entropy | $0.160 \pm 0.025$ | $0.153 \pm 0.022$ | $0.221 \pm 0.011$ | $0.179 \pm 0.018$ | $0.190 \pm 0.017$ |
| | VU | $0.160 \pm 0.022$ | $0.196 \pm 0.017$ | $0.223 \pm 0.014$ | $0.219 \pm 0.017$ | $0.188 \pm 0.020$ |
| | P(True) | $0.172 \pm 0.022$ | $0.215 \pm 0.017$ | $0.244 \pm 0.005$ | $0.215 \pm 0.015$ | $0.189 \pm 0.021$ |
| | LexSim | $0.151 \pm 0.024$ | $0.179 \pm 0.020$ | $0.225 \pm 0.010$ | $0.187 \pm 0.020$ | $0.174 \pm 0.019$ |
| | DegMat | $0.140 \pm 0.021$ | $0.164 \pm 0.018$ | $0.229 \pm 0.012$ | $0.191 \pm 0.018$ | $0.156 \pm 0.019$ |
| | LUQ | $0.148 \pm 0.020$ | 0.159 $\pm 0.016$ | $0.208 \pm 0.014$ | $0.180 \pm 0.019$ | **0.151** $\pm 0.019$ |
| | KLE | $0.188 \pm 0.021$ | $0.218 \pm 0.016$ | $0.244 \pm 0.006$ | $0.213 \pm 0.018$ | $0.193 \pm 0.021$ |
| | Inv-Entropy | 0.128 $\pm 0.020$ | **0.157** $\pm 0.018$ | **0.201** $\pm 0.014$ | **0.147** $\pm 0.017$ | 0.152 $\pm 0.020$ |
| | NI-Entropy | **0.124** $\pm 0.020$ | $0.164 \pm 0.017$ | 0.204 $\pm 0.014$ | 0.168 $\pm 0.021$ | $0.156 \pm 0.022$ |
| | NR-Inv-Entropy | $0.138 \pm 0.021$ | $0.159 \pm 0.015$ | $0.208 \pm 0.013$ | $0.188 \pm 0.021$ | $0.165 \pm 0.021$ |
| | WD-px-py | $0.184 \pm 0.019$ | $0.212 \pm 0.016$ | $0.225 \pm 0.010$ | $0.188 \pm 0.018$ | $0.169 \pm 0.021$ |
| | MAX-py-x | $0.148 \pm 0.019$ | $0.177 \pm 0.017$ | $0.229 \pm 0.011$ | $0.189 \pm 0.020$ | $0.169 \pm 0.018$ |

**Baselines** We compared our method with various state-of-the-art benchmarks highlighted in Sec. 1, Appendix A and a recent paper [Vashurin et al., 2024] identifying them as top-performers. These include: Semantic Entropy [Farquhar et al., 2024], Kernel Entropy [Gruber and Buettner, 2023], Verbalized Uncertainty (VU) [Tian et al., 2023], P(True) [Kadavath et al., 2022], Lexical Similarity (LexSim) [Fomicheva et al., 2020], Degree Matrix(DegMat) [Lin et al., 2024], Long-text Uncertainty Quantification (LUQ) [Zhang et al., 2024a], Kernel Language Entropy (KLE) [Nikitin et al., 2024]. We also include additional UQ measures based on our framework: (i) NI-Entropy: Non-inverse entropy which uses $P(Y \mid X)$ derived in Sec. 2.3 instead of $P(X \mid Y)$; the rest remains the same. (ii) NR-Inv-Entropy: entropy in (5) without replications. (iii) WD-px-py: Wasserstein distance $WD(P(X), P(Y))$ ; (iv) MAX-py-x: $\max_i P(y_i|x_i)$.

**Evaluation metrics** We evaluate performance using four metrics grouped into correctness-based and uncertainty-based categories. Correctness-based metrics: AUROC, PRR [Malinin and Gales, 2021], and Brier Score [Brier, 1950], measure how well confidence aligns with correctness. For MMLU, correctness is defined via exact match, while for other datasets we use GPT-3.5-Turbo to

assess whether a generated response is semantically equivalent to the reference (ground-truth) answer. Confidence is taken as the negative of the UQ measure (i.e. Inv-Entropy) for AUROC and PRR. For the Brier Score, we apply isotonic normalization [Zadrozny and Elkan, 2002] to map uncertainty scores to the [0,1] range. Correctness-based metrics, however, rely on ground truth and may fail in open-ended or weakly supervised settings. To address this, we propose the Temperature Sensitivity of Uncertainty (TSU), which quantifies how often uncertainty increases with temperature. Since higher temperatures flatten the softmax distribution, they should yield greater randomness and uncertainty [Hinton et al., 2015]. Formally, given a sequence of temperature values $\mathbb{T}_1 < \mathbb{T}_2 < \cdots < \mathbb{T}_n$, TSU is defined as:

$$\text{TSU}(\mathbb{T}_1, \mathbb{T}_2, \ldots, \mathbb{T}_n) = \frac{1}{|\mathcal{D}|} \sum_{S_x \in \mathcal{D}} \mathbb{I}\Big(\text{UQ}(S_x, \mathbb{T}_1) < \text{UQ}(S_x, \mathbb{T}_2) < \cdots < \text{UQ}(S_x, \mathbb{T}_n)\Big), \quad (7)$$

where $\mathcal{D}$ is the dataset, $S_x$ is a question in this dataset, $\text{UQ}(S_x, \mathbb{T})$ represents a UQ subroutine (such as Inv-Entropy) for input $S_x$ at temperature $\mathbb{T}$, and $\mathbb{I}(\cdot)$ is the indicator functionA salient feature in the definition of TSU in (7) is that it only depends on $S_x$, thus is agnostic to the "ground truth" output $y$. In addition, TSU extends beyond conventional correctness-based metrics by evaluating the granularity of uncertainty estimation. By leveraging temperature scaling as a probing mechanism, TSU assesses how effectively a method distinguishes between gradations of uncertainty.

**Implementation details**   Our framework requires three inputs: $\psi$, $\mathcal{P}er$, and $a_{\text{Similarity}}$. For $\mathcal{P}er$, we apply two strategies: (1) ChatGPT-based paraphrasing, generating nine perturbed versions per question, (2) GAAP introduced in Sec. 2.6 with a similarity threshold of $\delta = 0.7$. For $\psi$, we employ three state-of-the-art approaches: (i) SBERT-small (paraphrase-MiniLM-L6-v2), (ii) SBERT-large (all-mpnet-base-v2) [Reimers and Gurevych, 2019], and (iii) DeBERTa-v2-xlarge-MNLI [He et al., 2021]. For (i) and (ii), we use cosine similarity $a_{\text{Similarity}}(x, x') = (1 + \cos(x, x'))/2$. While, (iii) generates an entailment score; however, this score is not symmetric. To address this, we take the average $(a_{\text{Similarity}}(x, x') + a_{\text{Similarity}}(x', x))/2$. All experiments were conducted with an NVIDIA A100 GPU. Detailed experimental set up including prompts and parameters used are detailed in Appendix D. Appendix D also includes additional simulation results; we present only the core findings in the main paper for clarity and focus.

Table 2: Comparison of TSU across different temperature ranges for TriviaQA and MMLU.

| Method | TriviaQA | | | | MMLU | | | |
|---|---|---|---|---|---|---|---|---|
| | TSU(1.0,1.4) | TSU(0.7,1.4) | TSU(0.7–1.4) | TSU(0.3–1.4) | TSU(1.0,1.4) | TSU(0.7,1.4) | TSU(0.7–1.4) | TSU(0.3–1.4) |
| Semantic Entropy | 17.35 | 20.64 | 5.35 | 3.94 | 33.20 | 39.80 | 4.93 | 2.09 |
| Kernel Entropy | 43.92 | 55.56 | 18.23 | 9.64 | 59.48 | 69.10 | 37.32 | 12.63 |
| VU | 38.78 | 42.86 | 4.92 | 0.00 | 37.62 | 38.73 | 2.59 | 0.00 |
| P(True) | 3.85 | 3.49 | 0.00 | 0.00 | 5.87 | 5.79 | 0.00 | 0.00 |
| LexSim | 46.94 | 53.54 | 12.38 | 8.16 | 55.06 | 61.22 | 30.61 | 15.28 |
| DegMat | 45.37 | 47.96 | 20.02 | 13.27 | 69.39 | 77.55 | 32.58 | 14.34 |
| LUQ | 48.06 | 50.00 | 27.63 | 10.20 | 61.22 | 62.24 | 27.55 | 10.80 |
| KLE | 13.45 | 6.42 | 1.31 | 0.00 | 26.53 | 12.23 | 2.67 | 0.00 |
| Inv-Entropy | **77.55** | **88.78** | **47.21** | **19.05** | **73.47** | **86.73** | **43.88** | **18.37** |
| NI-Entropy | 61.22 | 60.20 | 19.32 | 11.73 | 50.00 | 59.38 | 18.37 | 7.14 |
| WD-px-py | 57.62 | 69.39 | 36.73 | 12.06 | 66.38 | 69.39 | 22.54 | 11.22 |
| MAX-py-x | 74.49 | 81.63 | 32.58 | 16.03 | 65.41 | 80.61 | 25.51 | 14.42 |

## 3.1   Results

**Correctness-based**   As shown in Table 1, Inv-Entropy achieves consistently strong and stable performance across all five datasets and all three metrics. It attains state-of-the-art results in both AUROC and PRR. The improvement is particularly clear on MMLU, where Inv-Entropy reaches 0.780 AUROC and 0.898 PRR, noticeably higher than all baselines, reflecting the advantage of our probabilistic framework with inverse design when the output information is limited. It also achieves leading performance on the long-answer dataset NQ, indicating that its effectiveness is not constrained by answer length (a detailed sensitivity analysis with respect to answer length is provided in Table 6 of the Appendix). It also ranks among the top two in Brier score, indicating well-calibrated confidence estimates. We intentionally use ChatGPT-based paraphrasing to highlight the advantages of our method independent of GAAP.

**TSU**   Table 2 reports TSU results TriviaQA and MMLU. These results align with the correctness-based UQ metrics in Table 1: methods defined by our probabilistic framework consistently achieve

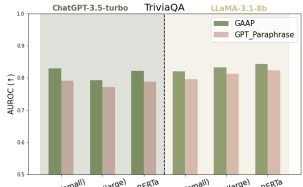 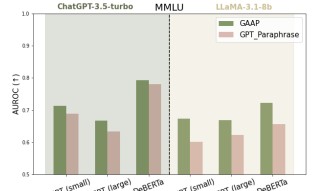 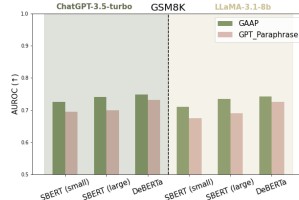

Figure 5: AUROC of Inv-Entropy under different perturbation methods (GAAP or ChatGPT-based paraphrasing) and embedding functions, on both ChatGPT and LLaMA models.

top performance, while LexSim, DegMat and LUQ exhibit highly inconsistent results, occasionally leading in specific cases but underperforming in others. Methods like P(True) perform poorly in TSU due to their binary nature, which fundamentally lacks granularity. Similarly, Semantic Entropy shows limited discriminatory power as its values often cluster around few discrete points. In contrast to these coarse-grained approaches, Inv-Entropy is fine-grained with superior ability to reflect uncertainty variations, demonstrated by consistent top TSU performance across all datasets and temperature settings. This result highlights the probabilistic framework's ability to capture uncertainty trends beyond labeled datasets. The complete TSU results are presented in Table 5 in the Appendix.

**Performance breakdown**    (i) *Impact of replications*: The comparison between NR-Inv-Entropy and Inv-Entropy in Tables 1 and 2 highlights the impact of replication and bootstrapping. Notably, NR-Inv-Entropy often performs competitively, showcasing the strength of our framework even without replication. This suggests that one can trade off a small loss in accuracy for fewer queries. Appendix D includes an ablation over $S$ and $r$, further confirming this finding. (ii) *Impact of inversion*: The comparison between NI-Entropy and Inv-Entropy further confirms the advantages of our inverse approach, which examines the diversity of inputs that could have led to a specific output. Although NI-Entropy consistently underperforms Inv-Entropy, it still often ranks second, highlighting the strength of our defined probabilistic framework even without the inversion. (iii) *Framework generality*: The often competitive performance of WD-px-py and MAX-py-x highlights the robustness of our framework and its flexibility in defining a wide range of UQ measures. (iv) *Impact of perturbations*: Fig. 5 shows that GAAP consistently improves AUROC across all three datasets and both LLMs compared to ChatGPT-based paraphrasing. By enhancing input diversity in a principled way, GAAP better tests model robustness and improves uncertainty quantification. These results underscore the importance of meaningful perturbations. (v) *Impact of embedding function*: Our framework delivers consistently strong uncertainty estimates with every encoder we tested, including SBERT (paraphrase-MiniLM-L6-v2), SBERT (all-mpnet-base-v2), and DeBERTa. Although the larger encoders provide slightly higher scores on several tasks, the overall gap is modest, showing that even lightweight models can support reliable UQ when paired with our method. The close alignment between results from SBERT (all-mpnet-base-v2) and DeBERTa further suggests that entailment and similarity signals extracted by the two architectures contain overlapping information.

## 4    Conclusion

We present a fully probabilistic framework for uncertainty quantification that models the conditional distribution of inputs given outputs through a dual random walk formulation. This inverse modeling perspective enables a principled characterization of uncertainty by capturing the semantic diversity of inputs associated with a given output. A key strength of our framework is its flexibility, allowing researchers to freely combine embedding functions, perturbation strategies, and similarity metrics to define customized uncertainty measures. As an instantiation of this idea, we introduce Inv-Entropy, a novel uncertainty metric derived from the framework. Together with the proposed perturbation algorithm GAAP and evaluation metric TSU, our method achieves state-of-the-art performance across multiple datasets. We believe this framework opens up broad opportunities for future research, providing a general foundation upon which new uncertainty measures, tailored to different purposes, can be systematically developed. We acknowledge that, like other perturbation and replication-based UQ methods, our approach may face practical limitations due to computational cost. A promising direction is to adaptively determine when further perturbation is unnecessary. We hope GAAP's sequential design can provide a step towards this goal.

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

**Appendix Outline**

# A    Related Work

UQ in LLMs has been gaining increasing interest recently [Huang et al., 2024]. The existing, albeit limited, literature can generally be categorized into three perspectives.

**Self-evaluation-based UQ:**    Self-evaluation techniques for LLMs [Chen et al., 2024] and the verbal expression of uncertainty [Lin et al., 2022] have recently been explored to enhance interpretability and reliability. For instance, recent approaches for evaluating free-form generation tasks frequently utilize LLMs as evaluators [Zheng et al., 2023]. Additionally, for gray-box models, where the internal workings are partially known, perplexity [Chen et al., 1998] and entropy [Shannon, 1948] can be directly computed from the output logits, providing a natural UQ measure.

**Replication-based UQ:**    These methods generate multiple outputs for a given input and measure the deviation between them to estimate uncertainty [Grewal et al., 2024, Wagner et al., 2024, Kuhn et al., 2023]. Perhaps most prevalent is semantic entropy [Farquhar et al., 2024], which computes uncertainty by clustering semantically equivalent answers from multiple responses and calculating the entropy of the resulting clusters. While effective in capturing aleatoric uncertainty, these approaches struggle with confidently wrong predictions, as resampling often yields similar incorrect results, leading to overconfidence and poor calibration. This issue exacerbates the challenges of handling hallucinations in LLMs [Grewal et al., 2024].

**Perturbation-based UQ:**    This is a more recent approach that involves systematic perturbations of inputs or latent representations to evaluate output variability [Zhang et al., 2023]. While Dong et al. [2023] systematically evaluates LLM robustness for noisy slot-filling tasks under diverse input perturbations, SPUQ [Gao et al., 2024] provides a UQ metric by analyzing response variations to perturbed inputs. Notably, SPUQ achieves significant improvements in model uncertainty calibration, reducing Expected Calibration Error (ECE) by an average of 50%. Indeed, our theoretical argument in Sec. 2.1 provides a theoretical justification for the need for perturbation in effective UQ.

In light of existing literature, our method unifies replication-based and perturbation-based UQ while introducing a Bayesian perspective to LLM uncertainty estimation by modeling the posterior distribution of inputs conditioned on outputs. This Bayesian inverse design approach provides a new framework and perspective for UQ of semantic models.

# B Proof of Lemma 2.1

In this section, we present the proof of Lemma 2.1. the proof is based on Taylor expansion of $\hat{f}(x')$.

The order-2 Taylor expansion for $\hat{f}(x_0 + (I - \eta_{\nabla f^\star})z)$ is,

$$\hat{f}(x_0 + (I - \eta_{\nabla f^\star})z) = \hat{f}(x_0) + \nabla\hat{f}(x_0)^\top(I - \eta_{\nabla f^\star})z + R(z), \tag{8}$$

where $R(z)$ is the remainder term defined as,

$$R(z) = \frac{1}{2}z^\top(I - \eta_{\nabla f^\star})\nabla^2\hat{f}(x_0 + \xi(z)(I - \eta_{\nabla f^\star})z)(I - \eta_{\nabla f^\star})z, \tag{9}$$

where $\xi(z) \in [0, 1]$ is a constant dependent on $z$.

By assumption, there exists a constant $G > 0$ such that $\|\nabla^2\hat{f}(x)\|_{op} \leq G, \forall x \in \mathbb{R}^d$. Therefore,

$$R(z) \leq G\|(I - \eta_{\nabla f^\star})z\|^2/2 \leq G\|z\|^2/2 \tag{10}$$

Since $z$ admits a $d$-dimensional isotropic Gaussian distribution $\mathcal{N}(0, \sigma^2)$, we can provide an upper bound for $\mathbb{E}(R(z))$ and $\mathbb{E}(R^2(z))$ as

$$\mathbb{E}(R(z)) \leq Gd\sigma^2/2, \tag{11}$$

and

$$\mathbb{E}(R(z)^2) \leq \frac{G^2}{4}\mathbb{E}[\|z\|^4]$$
$$\leq \frac{G^2}{4}d^2\mathbb{E}[z_i^4] = \frac{G^2}{4}d^2 3\sigma^4. \tag{12}$$

We can also calculate the expectation of $\hat{f}(x_0 + (I - \eta_{\nabla f^\star})z)$ as

$$\mathbb{E}[\hat{f}(x_0 + (I - \eta_{\nabla f^\star})z)] = \hat{f}(x_0) + \mathbb{E}(R(z)). \tag{13}$$

Then, the variance of $\hat{f}(x_0 + (I - \eta_{\nabla f^\star})z)$ is,

$$\mathbb{V}[\hat{f}(x_0 + (I - \eta_{\nabla f^\star})z)] = \mathbb{E}\left[\left(\hat{f}(x_0 + (I - \eta_{\nabla f^\star})z) - \mathbb{E}[\hat{f}(x_0 + (I - \eta_{\nabla f^\star})z)]\right)^2\right]$$
$$= \mathbb{E}\left[\left(\nabla\hat{f}(x_0)^\top(I - \eta_{\nabla f^\star})z\right)^2\right] + 2\mathbb{E}\left[\left(\nabla\hat{f}(x_0)^\top(I - \eta_{\nabla f^\star})z\right)\zeta(z)\right] + \mathbb{E}\left[\zeta(z)^2\right], \tag{14}$$

where $\zeta(z)$ is defined as

$$\zeta(z) = R(z) - \mathbb{E}(R(z)). \tag{15}$$

Notice that the first term in (14) is,

$$\mathbb{E}\left[\left(\nabla\hat{f}(x_0)^\top(I - \eta_{\nabla f^\star})z\right)^2\right]$$
$$= \mathbb{E}\left[\nabla\hat{f}(x_0)^\top(I - \eta_{\nabla f^\star})zz^\top(I - \eta_{\nabla f^\star})\nabla\hat{f}(x_0)\right]$$
$$= \sigma^2\nabla\hat{f}(x_0)^\top(I - \eta_{\nabla f^\star})\nabla\hat{f}(x_0)$$
$$= \sigma^2\left(\|\nabla\hat{f}(x_0)\|^2 - \frac{\left(\nabla f^\star(x_0)^\top\nabla\hat{f}(x_0)\right)^2}{\|\nabla f^\star(x_0)\|^2}\right) \tag{16}$$
$$= \sigma^2\|\nabla\hat{f}(x_0)\|^2\sin^2\theta(\nabla f^\star(x_0), \nabla\hat{f}(x_0)).$$

The third term in (14) is upper bounded by

$$\mathbb{E}\left[\zeta(z)^2\right]$$
$$\leq 2\mathbb{E}[R(z)^2] + 2\mathbb{E}[R(z)]^2 \tag{17}$$
$$\leq 2G^2d^2\sigma^4 = O(\sigma^4),$$

where we used (11) and (12) in the third inequality.

The second term in (14) is upper bounded by the Holder inequality,

$$
\begin{aligned}
&\mathbb{E}\left[\left(\nabla \hat{f}(x_0)^\top (I - \eta_{\nabla f^\star})z\right)\zeta(z)\right] \\
&\leq \sqrt{\mathbb{E}\left[\left(\nabla \hat{f}(x_0)^\top (I - \eta_{\nabla f^\star})z\right)^2\right]}\sqrt{\mathbb{E}\left[\zeta(z)^2\right]} \\
&\leq \sigma \|\nabla \hat{f}(x_0)\| \sin\theta(\nabla f^\star(x_0), \nabla \hat{f}(x_0))\sqrt{2}Gd\sigma^2 \\
&= O(\sigma^3),
\end{aligned}
\tag{18}
$$

where we used the inequality (17) in the second inequality, and the assumption that $\|\nabla \hat{f}(x_0)\|$ is upper bounded in the last theorem.

We complete the proof by combining (16), (18), and (17).

## C GAAP Implementation Details

In this section, we first introduce an example of the perturbation set $\mathcal{P}er(S_0)$, then introduce the details of our GAAP algorithm that is used for the perturbation function $\mathcal{P}er(\cdot)$.

### C.1 An example

We show an example of $\mathcal{P}er(S_{x_0})$ in Table 3. The original input is a question from TriviaQA $S_{x_0} =$"Which golfer became only the fifth in history to get both the British and US Open championships in the same year, in 1982? ".

| $S_{x_0}$ | Which golfer became only the fifth in history to get both the British and US Open championships in the same year, in 1982? |
|---|---|
| $S_{x_1}$ | Which golfer became only the fifth in history to get both the British and US Open in the same year, in 1982? |
| $S_{x_2}$ | Which driver became only the fifth in history to get both the British and US Open triple_crown in the same year, in 1982? |
| $S_{x_3}$ | Which medalist became only the fifth in history to get both the British and US Open high_status in the same year, in 1982? |
| $S_{x_4}$ | Which golfer became only the fifth in history to win both the British and US Open championships in the same year, in 1982? |
| $S_{x_5}$ | Which golfer became only the fifth in history to win both the British and US Open in the same year, in 1982? |
| $S_{x_6}$ | Which driver became only the fifth in history to win both the British and US Open triple_crown in the same year, in 1982? |
| $S_{x_7}$ | Which medalist became only the fifth in history to win both the British and US Open title in the same year, in 1982? |
| $S_{x_8}$ | Which driver became only the fifth in history to win both the British and US Open championships in the same year, in 1982? |
| $S_{x_9}$ | Which driver became only the fifth in history to win both the British and US Open in the same year, in 1982? |
| $S_{x_{10}}$ | Which linksman became only the fifth in history to win both the British and US Open in the same year, in 1982? |

Table 3: A question $S_{x_0}$ and 10 perturbed versions of $S_{x_0}$ as the output of GAAP.

### C.2 Algorithm details

We first introduce some notations. For an input sentence $S_{x_0}$, we can denote it in the form of a token (word) series,

$$
S_{x_0} = (t_1, t_2, \ldots, t_p),
$$

where $t_1, t_2, \ldots, t_p$ denote the sequence of tokens that constitute the input text $S_{x_0}$, arranged in their original order. We will then elaborate on each step of GAAP.

#### C.2.1 Key words selection

For better sampling efficiency, GAAP does not perturb all tokens equally. Instead, we identify the key tokens in $S_{x_0}$ first and only perturb these tokens. As a result, we could explore the semantic space more efficiently under the perturbation budget constraint.

We define a function $k(\cdot, \cdot)$ that identifies key tokens within a given text. The function takes two inputs: the first is a text and the second is a ratio indicating the proportion of key tokens to all tokens in this text. $k(S_{x_0}, r)$ returns a subset of tokens:

$$
k(S_{x_0}, r) = \{t_{j_1}, t_{j_2}, \ldots, t_{j_q}\},
$$

where the indices $\{j_1, j_2, \ldots, j_q\} \subseteq \{1, \ldots, p\}$, and the number of selected key tokens are: $q = \text{int}[r \cdot p]$. In GAAP, we use KeyBERT [Grootendorst, 2024] to implement $k(\cdot, \cdot)$.

### C.2.2 Initial population generation

Next, we define a replacement function $re(\cdot, \cdot, \cdot)$ that substitutes a specific token in the sequence. The function is defined as follows:

$$re(S_{x_0}, t_{j_i}, t'_{j_i}) = (t_1, \ldots, t_{j_i-1}, t'_{j_i}, t_{j_i+1}, \ldots, t_p). \tag{19}$$

In GAAP, we choose $t'_{j_i}$ from a substitution set $\mathrm{SUB}(t_{j_i})$. And the substitution set is defined as the union of all possible hypernyms, hyponyms, synonyms, and an empty set denoting word deletion, $\mathrm{SUB}(t_{j_i}) = \mathrm{hypernyms}(t_{j_i}) \cup \mathrm{hyponyms}(t_{j_i}) \cup \mathrm{synonyms}(t_{j_i}) \cup \{\emptyset\}$.

The initial population of GAAP for the input $S_{x_0}$ is defined as the union of the outcomes of all possible single-key token perturbations,

$$\mathrm{Pop}_0(S_{x_0}) = \bigcup_{t_{j_i} \in k(S_{x_0}, r)} \bigcup_{t'_{j_i} \in \mathrm{SUB}(t_{j_i})} re(S_{x_0}, t_{j_i}, t'_{j_i}).$$

### C.2.3 Iterative population update

Then, we introduce an iterative scheme for the perturbed population to evolve. In each iteration, we must follow the next three steps in sequence.

1. **Selection**: The selection step aims to choose a subset of individuals from the population as parents for subsequent procedures. In GAAP, we design a random selection mechanism where individuals whose semantic meanings are closer to those of the original text $S_{x_0}$ will be selected with higher probability.

   In the terminology of genetic algorithms, we define our fitness function as the semantic similarity to $x_0$, $a_{\mathrm{Similarity}}(x_0, \cdot)$. Then, we compute the fitness value $a_{\mathrm{Similarity}}(x_0, x_i)$ for $\forall S_{x_i} \in \mathrm{Pop}_t(S_{x_0})$, and use roulette wheel selection [Lipowski and Lipowska, 2012] to choose parents. More specifically, the probability of selecting $S_{x_i}$ is:

   $$P(S_{x_i}) = \frac{a_{\mathrm{Similarity}}(x_0, x_i)}{\sum_{S_{x_j} \in \mathrm{Pop}_t(S_{x_0})} a_{\mathrm{Similarity}}(x_0, x_j)}$$

   The set of all selected parent individuals is denoted as $\mathrm{Pa}_t(S_{x_0})$.

2. **Crossover**: The crossover step aims to generate new offspring by recombining the segments (i.e., token sub-sequences) of parent individuals.

   The inputs of the crossover operation are two randomly selected parent individuals $S_{x_A}$ and $S_{x_B}$ from $\mathrm{Pa}_t(S_{x_0})$. Then, we uniformly randomly sample a crossover point $h$ from $\{1, 2, \ldots, p-1\}$), where $p$ is the length of the shorter one between $S_{x_A}$ and $S_{x_B}$. Next, we generate two offspring individuals $S_{x_{A'}}$ and $S_{x_{B'}}$ as

   $$S_{x_{A'}} = (t_1^A, \ldots, t_h^A, t_{h+1}^B, \ldots, t_p^B),$$
   $$S_{x_{B'}} = (t_1^B, \ldots, t_h^B, t_{h+1}^A, \ldots, t_p^A).$$

   where $t_i^A$ and $t_i^B$ represent the $i$-th token of parents $S_{x_A}$ and $S_{x_B}$, respectively. The set of all generated offspring individuals is denoted as $\mathrm{Off}_t(S_{x_0})$.

3. **Mutation**: The mutation operation aims to augment population diversity again by randomly replacing certain tokens in the offspring individuals. For each offspring individual $S_{x_i} \in \mathrm{Off}_t(S_{x_0})$, we randomly select a token $t_{j_i}$ for replacement. Then, we uniformly randomly choose a new token $t'_{j_i}$ from the substitution set $\mathrm{SUB}(t_{j_i})$:

   $$t'_{j_i} \in \mathrm{SUB}(t_{j_i}) = \mathrm{hypernyms}(t_{j_i}) \cup \mathrm{hyponyms}(t_{j_i}) \cup \mathrm{synonyms}(t_{j_i}) \cup \{\emptyset\}.$$

   Finally, we generate the mutated individual as

   $$S_{x'_i} = re(S_{x_i}, t_{j_i}, t'_{j_i}),$$

   where $re(\cdot, \cdot, \cdot)$ is the replacement function defined in (19). The set of all mutated offspring individuals is denoted as $\mathrm{Mu}_t(S_{x_0})$.

The new population is formed by combining all the 3 sets above,

$$\text{Pop}_{t+1}(S_{x_0}) = \text{Pa}_t(S_{x_0}) \cup \text{Off}_t(S_{x_0}) \cup \text{Mu}_t(S_{x_0}). \tag{20}$$

Equation (20) defines an iterative algorithm to update the population for $t = 0, 1, 2, \cdots$. The iterative process terminates when either of the following conditions is met: (1) The number of generations $t$ exceeds a predefined maximum value $Num$: $t \geq Num$, (2) the maximum of fitness values in the population is smaller than a threshold $\delta$: $\max_i a_{\text{Similarity}}(x_0, x_i) < \delta$.

### C.2.4 Perturbation set construction

Unlike standard genetic algorithms that aim to optimize the fitness function to its extreme value, the objective for GAAP is to use populations $\text{Pop}_t$ at different generations $t$ to construct a perturbation set $\mathcal{P}er(S_{x_0})$. The ideal perturbation set should be diverse and contain texts with varying degrees of similarity to $S_{x_0}$. Since earlier generations contain fewer perturbations and later generations involve more perturbations, the generation index $t$ is a natural indicator of similarity. As a result, we generate $\mathcal{P}er(S_{x_0})$ by random sampling from populations at different generations.

More specifically, $\mathcal{P}er(S_{x_0})$ consists of random samples from populations at generations $t = 0, \tau, 2\tau, ...$, where $\tau \in \mathbb{Z}$ is the sample interval:

$$\mathcal{P}er(S_{x_0}) = \bigcup_{q=0}^{\text{int}\left(\frac{T}{\tau}\right)} \text{Uniform}(\text{Pop}_{q\tau}(S_{x_0})) \tag{21}$$

where $\text{Uniform}(\cdot)$ is a function that uniformly randomly selects a subset of individuals from the population. The construction rule (21) ensures that $\mathcal{P}er(S_{x_0})$ contains texts with progressively decreasing similarity to $S_{x_0}$, as the genetic algorithm evolves toward lower fitness values (i.e., lower similarity).

After the termination condition is met, the perturbation set $\mathcal{P}er(S_{x_0})$ is returned as the final output. This set represents a collection of perturbed versions of $S_{x_0}$, each with a different degree of perturbation.

## D Settings & Additional Experimental Results

### D.1 Experimental setting

**Dataset Preprocessing**  No preprocessing was required except for MMLU, due to its heterogeneous and often non-standard question formats, which are incompatible with our perturbation-based framework. We excluded items that lacked self-contained semantic meaning and thus were unsuitable for perturbation (e.g., "Which of the following statements is true?"), purely mathematical expressions without contextual meaning (e.g., "If $A = (1, 2, 3, 4)$, let $B = \{(1, 2), (1, 3), (4, 2)\}$. Then $B$ is"), or other irregular or ambiguous phrasing. Only questions with clearly worded, self-contained statements were retained.

The following are all the prompts used in our experiments.

**ChatGPT-based paraphrasing**

```
Please Provide {number of perturbations} paraphrases for this sentence:
{sentence}
```

**Generating Responses**

For TriviaQA, SciQ, and NQ:

```
{question} Answer concisely and return only the name.
```

For MMLU:

```
{question + choices} Answer concisely and return only the name.
```

For GSM8K:

```
{question} Answer concisely and return only the result itself.
```

We design our prompts to align closely with the highly concise reference answers in the datasets. The same prompts are used for both our method and all baseline models, across both GPT-3.5-Turbo and LLaMA-3.1-8B-Instruct.

### Correctness Evaluation

```
Are the following two answers to my question Q semantically equivalent?
Q: {question}
A1: {standard answer}
A2: {answer}
Please answer with a single word, either Yes or No.
```

### VU Derivation (used as one of our benchmarks)

After the previous prompt of generating responses, we append the following prompt to elicit verbalized uncertainty:

```
And use a percentage to tell me your confidence in your answer.
```

The parameters used in our experiments are listed below.

### Perturbation Configuration

As detailed in the main manuscript, we generate nine perturbations per question using ChatGPT-based paraphrasing, resulting in ten variants per question including the original. For GAAP, we set the threshold $\delta = 0.7$ and also fix the number of perturbations at nine. This uniformity is crucial for our probabilistic framework, where each question induces a distribution over variants. To ensure that these distributions are comparable and defined on the same scale, we require the same number of perturbations per question across the dataset. When fewer than nine are generated, we randomly duplicate existing perturbations; when more than nine are produced, we randomly sample nine.

### Replication and Bootstrapping

Our model incorporates bootstrapping to utilize replicated responses. Unless otherwise specified (notably in the ablation studies analyzing the impact of $S$ and $r$), all reported results are based on experiments with $S = 30$ bootstrapping iterations and $r = 5$ replications.

### Calculation of Mean and Variance

All reported evaluation metrics represent means with associated standard deviations computed via bootstrapping. Using 40 bootstrap samples generated with replacement, we: (i) Calculate the target metric for each sample, (ii) Aggregate results by taking the mean of sample-level metrics as the final estimate, (iii) Compute the standard deviation across the 40 values as a dispersion measure.

### LLM Configuration

For ChatGPT-based paraphrasing, we set the temperature to 0.7. For correctness evaluation, a temperature of 0 is used. For LLaMA, due to LLaMA's lack of automatic response termination and occasional output corruption, this may lead to incomplete or malformed answers. To mitigate this, we adopt a multi-attempt generation protocol with the following cleaning steps to ensure concise and valid outputs: (i) remove the echoed question if present, (ii) delete formatting tokens (e.g., [INST], [/INST], #) and any trailing text, and (iii) retain only the first non-empty line after trimming whitespace. The following is an example.

Question:

```
What is the capital of France?
```

Response before cleaning:

```
What is the capital of France?
Paris [/INST]#
It is a major European city and a global
```

Response after cleaning:

```
Paris
```

## D.2 Computational cost comparison

For non-locally hosted LLMs, uncertainty quantification methods generally involve two stages: obtaining responses from the model via API calls and computing the uncertainty scores based on those responses. This applies to both our method and all baselines; the only exception is Verbalized Uncertainty (VU), which directly returns a score from the API without requiring post-processing. As discussed in the paper, any method that relies on perturbations and/or replications incurs additional computational cost. However, a major advantage is that responses for perturbed inputs can be generated in parallel, making the process more scalable in practice.

With this in mind, we divide the computational cost into two components:

  (A) Computation time after responses are collected, and

  (B) Total cost, which includes (i) along with API call time and perturbation generation.

**Regarding API cost:** Our method introduces $n$ perturbations per question and generates $r$ response replications for each version, resulting in a total of $(n + 1) \times r$ API calls per input. Thus, the cost scales linearly with both the number of perturbations and replications. That said, while our method introduces perturbations, it requires far fewer replications than several existing baselines. For instance, Semantic Entropy relies heavily on large $r$ values (with $n = 0$); prior work recommends $r > 10$ for stable performance [Farquhar et al., 2024] In contrast, we demonstrate that even without replication ($r = 1$), our method remains effective. This is evidenced by the strong performance of **NR-Inv-Entropy**, which uses perturbation alone and still consistently outperforms many baselines.

**Regarding computation after responses are collected:** Below, we present the compute results in Table 4 without parallelization. The reported numbers are averaged over all sampled questions in TriviaQA. As shown, **our method is highly efficient in the post-processing stage** when computing Inv-Entropy scores. Furthermore, in settings where response generation time is negligible, such as when LLMs are deployed locally and perturbations are processed in parallel, our total computational cost is often lower than that of existing baselines.

Table 4: Computation efficiency comparison across methods.

| Method | Peak GPU Memory (MB) | Time A (s) | Time B (s) |
|---|---|---|---|
| Semantic Entropy | 3575.31 | 13.985 | 20.183 |
| VU | 0 | 0 | 0.620 |
| P(True) | 0 | 0.001 | 2.201 |
| LexSim | 0 | 0.022 | 3.189 |
| DegMat | 1610.29 | 6.143 | 15.905 |
| LUQ | 1580.21 | 3.821 | 4.432 |
| KLE | 1608.52 | 1.323 | 6.725 |
| Inv-Entropy (Ours) | 86.65 | 1.990 | 10.323 |
| NR-Inv-Entropy (Ours) | 86.65 | 1.769 | 6.358 |

## D.3 Additional experimental results

Table 5 presents the complete TSU results across all five datasets.

Table 5: Comparison of average TSU values across all five datasets and temperatures $\mathbb{T} = \{0.3, 0.7, 1.0, 1.4\}$, using GPT-3.5-Turbo with ChatGPT-based paraphrasing and SBERT-small (paraphrase-MiniLM-L6-v2) embeddings. $\text{TSU}(a, b, \ldots, c)$ is abbreviated as $\text{TSU}(a\text{--}c)$ (e.g., $\text{TSU}(0.3, 0.7, 1.0)$ becomes $\text{TSU}(0.3\text{--}1.0)$). All values are reported as percentages.

| Method | TSU (0.3, 0.7) | TSU (0.7, 1.0) | TSU (1.0, 1.4) | TSU (0.3, 1.0) | TSU (0.3, 1.4) | TSU (0.7, 1.4) | TSU (0.3-1.0) | TSU (0.7-1.4) | TSU (0.3-1.4) |
|---|---|---|---|---|---|---|---|---|---|
| **TriviaQA** | | | | | | | | | |
| Semantic Entropy | 11.57 | 14.26 | 17.35 | 21.56 | 25.51 | 20.64 | 5.18 | 5.35 | 3.94 |
| VU | 22.45 | 27.55 | 38.78 | 25.51 | 41.84 | 42.86 | 0.00 | 4.92 | 0.00 |
| P(True) | 1.02 | 2.13 | 3.85 | 1.38 | 0.98 | 3.49 | 0.00 | 0.00 | 0.00 |
| LexSim | 22.39 | 22.95 | 46.94 | 30.61 | 53.36 | 53.54 | 9.18 | 12.38 | 8.16 |
| DegMat | 24.58 | 33.21 | 45.37 | 31.77 | 48.98 | 47.96 | 18.37 | 20.02 | 13.27 |
| LUQ | 20.41 | 31.08 | 48.06 | 33.67 | 52.34 | 50.00 | 14.78 | 27.63 | 10.20 |
| KLE | 7.14 | 17.35 | 13.45 | 4.57 | 1.28 | 6.42 | 2.79 | 1.31 | 0.00 |
| Inv-Entropy | 52.42 | 66.33 | **77.55** | 76.53 | 92.86 | **88.78** | 30.49 | **47.21** | **19.05** |
| NI-Entropy | 55.14 | 47.96 | 61.22 | 59.35 | 67.35 | 60.20 | 21.43 | 19.32 | 11.73 |
| WD-px-py | 44.81 | **67.35** | 57.62 | 60.20 | 64.49 | 69.39 | 23.11 | 36.73 | 12.06 |
| MAX-py-x | **65.31** | 50.83 | 74.49 | **67.35** | 85.71 | 81.63 | 27.42 | 32.65 | 16.03 |
| **SciQ** | | | | | | | | | |
| Semantic Entropy | 11.09 | 15.48 | 21.55 | 17.35 | 27.62 | 25.51 | 6.73 | 4.31 | 2.65 |
| VU | 21.43 | 33.67 | 28.57 | 35.71 | 31.42 | 32.65 | 1.55 | 6.12 | 0.00 |
| P(True) | 0.00 | 3.42 | 3.14 | 0.00 | 1.33 | 0.79 | 0.00 | 0.00 | 0.00 |
| LexSim | 31.73 | 29.49 | 47.96 | 40.82 | 61.22 | 59.18 | 19.39 | 13.27 | 11.22 |
| DegMat | 39.80 | 47.96 | 57.14 | 52.04 | 64.28 | 64.46 | 23.47 | 23.99 | 13.04 |
| LUQ | 38.45 | 43.88 | 41.84 | 44.90 | 57.14 | 44.95 | 18.37 | 17.35 | 6.12 |
| KLE | 6.12 | 39.54 | 19.32 | 4.98 | 5.67 | 18.37 | 2.30 | 5.10 | 0.00 |
| Inv-Entropy | 46.70 | **62.24** | **72.38** | **66.72** | **75.84** | **80.61** | 22.45 | **35.71** | 14.31 |
| NI-Entropy | 42.30 | 59.18 | 55.19 | 48.98 | 54.77 | 63.27 | 21.43 | 22.65 | 9.18 |
| WD-px-py | 50.00 | 59.16 | 63.25 | 59.37 | 71.40 | 69.39 | 22.13 | 28.59 | 11.22 |
| MAX-py-x | **51.02** | 61.22 | 67.35 | 61.22 | 69.86 | 73.25 | **24.73** | 34.69 | **15.29** |
| **NQ** | | | | | | | | | |
| Semantic Entropy | 22.53 | 20.41 | 35.71 | 27.55 | 39.80 | 30.78 | 4.08 | 10.20 | 2.04 |
| VU | 23.71 | 22.68 | 40.21 | 21.65 | 36.08 | 36.08 | 4.12 | 3.09 | 0.00 |
| P(True) | 4.08 | 3.06 | 2.04 | 1.02 | 2.04 | 1.02 | 0.00 | 0.00 | 0.00 |
| LexSim | 51.55 | 54.64 | 59.79 | 63.73 | 79.38 | 74.23 | **36.08** | 32.99 | 17.44 |
| DegMat | 51.04 | 51.04 | 59.38 | 65.62 | 76.04 | 71.88 | 32.29 | 33.33 | 21.42 |
| LUQ | 43.16 | 49.47 | 67.37 | 57.89 | 76.84 | 74.74 | 24.21 | 30.41 | 13.68 |
| KLE | 6.38 | 28.72 | 21.28 | 6.38 | 4.26 | 12.77 | 2.13 | 4.26 | 0.00 |
| Inv-Entropy | 61.86 | 59.79 | **78.35** | 74.23 | **91.75** | **85.57** | 30.93 | **41.24** | **22.43** |
| NI-Entropy | 47.42 | 60.82 | 59.79 | 58.76 | 70.10 | 70.10 | 23.71 | 29.81 | 10.31 |
| WD-px-py | **63.55** | 50.52 | 70.10 | 67.01 | 76.29 | 72.16 | 27.84 | 25.77 | 16.92 |
| MAX-py-x | 55.67 | **62.89** | 73.20 | 66.30 | 84.54 | 82.47 | 28.87 | 39.18 | 18.56 |
| **MMLU** | | | | | | | | | |
| Semantic Entropy | 21.43 | 17.35 | 33.20 | 21.83 | 42.08 | 39.80 | 7.14 | 4.93 | 2.09 |
| VU | 23.56 | 19.01 | 37.62 | 30.18 | 32.50 | 38.73 | 1.37 | 2.59 | 0.00 |
| P(True) | 1.92 | 4.56 | 5.87 | 4.92 | 5.02 | 5.79 | 0.00 | 0.00 | 0.00 |
| LexSim | 33.94 | 41.17 | 55.06 | 50.00 | 68.37 | 61.22 | 24.78 | 30.61 | 15.28 |
| DegMat | 41.84 | 54.08 | 69.39 | 53.76 | 78.57 | 77.55 | 21.46 | 32.58 | 14.34 |
| LUQ | 53.46 | 47.96 | 61.22 | 58.92 | 68.37 | 62.24 | 27.55 | 27.55 | 10.80 |
| KLE | 10.20 | 25.51 | 26.53 | 7.14 | 4.76 | 12.23 | 2.93 | 2.67 | 0.00 |
| Inv-Entropy | 60.08 | **67.35** | **73.47** | **79.59** | **90.82** | **86.73** | **34.31** | **43.88** | **18.37** |
| NI-Entropy | 56.12 | 54.63 | 50.00 | 67.35 | 52.45 | 59.38 | 19.39 | 18.37 | 7.14 |
| WD-px-py | 50.00 | 59.16 | 66.38 | 59.37 | 70.41 | 69.39 | 25.60 | 21.65 | 11.22 |
| MAX-py-x | **62.24** | 62.35 | 65.41 | 73.47 | 81.73 | 80.61 | 31.62 | 25.51 | 14.42 |
| **GSM8K** | | | | | | | | | |
| Semantic Entropy | 44.90 | 56.12 | 35.71 | 71.43 | 77.55 | 62.24 | 20.41 | 13.27 | 4.08 |
| VU | 11.34 | 39.18 | 29.90 | 35.05 | 35.05 | 38.14 | 2.06 | 6.19 | 1.03 |
| P(True) | 5.10 | 17.35 | 6.12 | 3.32 | 3.98 | 11.22 | 0.00 | 0.00 | 0.00 |
| LexSim | 54.17 | 63.54 | 54.17 | 65.62 | 64.58 | 63.54 | 30.31 | 23.96 | 10.42 |
| DegMat | 55.79 | 64.21 | 55.79 | 72.63 | 75.79 | 70.53 | 29.47 | 29.47 | 10.53 |
| LUQ | 64.95 | **74.23** | 72.16 | 83.51 | 89.69 | **81.44** | 44.33 | **54.64** | **37.11** |
| KLE | 17.02 | 35.11 | 27.66 | 4.23 | 3.72 | 23.40 | 0.00 | 3.19 | 0.00 |
| Inv-Entropy | **73.68** | 67.33 | 68.42 | 86.32 | **93.68** | 72.63 | **45.81** | 44.21 | 30.53 |
| NI-Entropy | 53.61 | 58.76 | 51.55 | 61.16 | 57.73 | 59.79 | 25.77 | 20.62 | 6.19 |
| WD-px-py | 54.08 | 53.06 | 68.37 | 55.10 | 64.29 | 60.20 | 20.41 | 27.43 | 7.14 |
| MAX-py-x | 71.88 | 65.62 | 62.50 | **89.58** | 91.67 | 77.08 | 42.71 | 37.50 | 28.12 |

Table 6 shows the performance across different answer lengths in Natural Question (NQ) [Kwiatkowski et al., 2019] dataset. NQ dataset includes both short and long-form answers. We selected questions that contain only long-form answers. The reference answers in this subset range from 34 to 350 tokens in length. To analyze the sensitivity of different answer lengths, we further divided the dataset into three subsets based on the number of tokens in the reference answers: Short (< 80 tokens), Medium (80–120 tokens) and Long ($\geq$ 120 tokens). We evaluated both baseline methods and our proposed method on each of these subsets as well as the full sample set.

Our method achieves state-of-the-art performance on the full dataset, ranking first across all three metrics. We also observe some sensitivity to answer length: it shows a clear and substantial advantage on the short-answer subset, highlighting the effectiveness of the inverse-design mechanism when responses are more concise. On the medium-length subset, it continues to outperform all baselines, though with a smaller margin. On the long-answer subset, while not the top performer, our method remains competitive and yields reasonable results compared to strong baselines such as LUQ.

Intuitively, these results align with expectations under the inverse perspective: the shorter the answer, the more important it becomes to explore the diversity of the input space. Nevertheless, our approach consistently ranks among the top two methods even in the long-form QA setting. We end by noting that our probabilistic framework is also generic by nature and can accommodate a forward perspective or alternative uncertainty metrics beyond entropy computed over $P(X \mid Y)$. Exploring such metrics across different answer lengths is a promising direction we leave for future work.

Table 6: Comparison of AUROC, PRR, and Brier scores across Short, Medium, Long, and Full subsets on NQ. **Bold** indicates the best performer. Underline indicates the second-best.

| Metric (↑/↓) | Method | Short | Medium | Long | Full |
|---|---|---|---|---|---|
| **AUROC (↑)** | Semantic Entropy | 0.509 | 0.461 | 0.584 | 0.521 |
| | VU | 0.531 | 0.495 | 0.508 | 0.533 |
| | P(True) | 0.529 | 0.473 | 0.548 | 0.519 |
| | LexSim | 0.624 | 0.438 | 0.555 | 0.518 |
| | DegMat | 0.547 | 0.621 | 0.484 | 0.551 |
| | LUQ | 0.662 | 0.508 | 0.612 | 0.627 |
| | KLE | 0.265 | 0.456 | 0.445 | 0.410 |
| | Inv-Entropy (Ours) | **0.794** | **0.634** | 0.589 | **0.661** |
| **PRR (↑)** | Semantic Entropy | 0.420 | 0.584 | 0.507 | 0.505 |
| | VU | 0.489 | 0.615 | 0.495 | 0.537 |
| | P(True) | 0.427 | 0.584 | 0.478 | 0.502 |
| | LexSim | 0.628 | 0.684 | 0.544 | 0.563 |
| | DegMat | 0.543 | 0.682 | **0.548** | 0.549 |
| | LUQ | 0.649 | 0.655 | 0.523 | 0.595 |
| | KLE | 0.354 | 0.649 | 0.444 | 0.449 |
| | Inv-Entropy (Ours) | **0.747** | **0.742** | 0.508 | **0.614** |
| **Brier (↓)** | Semantic Entropy | 0.227 | 0.230 | 0.221 | 0.242 |
| | VU | 0.209 | 0.218 | 0.216 | 0.223 |
| | P(True) | 0.225 | 0.232 | 0.230 | 0.244 |
| | LexSim | 0.169 | 0.199 | 0.207 | 0.225 |
| | DegMat | 0.187 | 0.181 | 0.213 | 0.229 |
| | LUQ | 0.164 | 0.209 | **0.179** | 0.208 |
| | KLE | 0.228 | 0.203 | 0.225 | 0.244 |
| | Inv-Entropy (Ours) | **0.125** | **0.175** | 0.193 | **0.201** |

Table 7 shows how the experimental results vary with the number of bootstrapping iterations $S$. We observe that increasing $S$ initially enhances the performance of our method, Inv-Entropy. However, beyond a certain threshold, further increases yield diminishing or no significant returns. Therefore, using more than 10 iterations appears to provide limited additional benefit.

Table 7: Comparison of Inv-Entropy performance across all the 5 datasets with varying numbers of bootstrapping iterations $S$. We use GPT-3.5-Turbo with ChatGPT-based paraphrasing and DeBERTa-v2-xlarge-MNLI embedding function.

| Dataset | Metric | $S=1$ | $S=5$ | $S=10$ | $S=30$ | $S=50$ | $S=100$ |
|---|---|---|---|---|---|---|---|
| TriviaQA | AUROC | $0.743_{\pm 0.061}$ | $0.781_{\pm 0.057}$ | $0.785_{\pm 0.056}$ | $0.788_{\pm 0.054}$ | $0.780_{\pm 0.054}$ | $0.788_{\pm 0.054}$ |
| | PRR | $0.840_{\pm 0.054}$ | $0.881_{\pm 0.043}$ | $0.882_{\pm 0.043}$ | $0.885_{\pm 0.044}$ | $0.882_{\pm 0.044}$ | $0.885_{\pm 0.044}$ |
| | Brier | $0.138_{\pm 0.021}$ | $0.127_{\pm 0.020}$ | $0.125_{\pm 0.020}$ | $0.128_{\pm 0.020}$ | $0.131_{\pm 0.021}$ | $0.128_{\pm 0.020}$ |
| SciQ | AUROC | $0.724_{\pm 0.052}$ | $0.733_{\pm 0.050}$ | $0.740_{\pm 0.049}$ | $0.740_{\pm 0.050}$ | $0.739_{\pm 0.050}$ | $0.743_{\pm 0.049}$ |
| | PRR | $0.821_{\pm 0.052}$ | $0.840_{\pm 0.046}$ | $0.844_{\pm 0.045}$ | $0.843_{\pm 0.042}$ | $0.842_{\pm 0.046}$ | $0.845_{\pm 0.046}$ |
| | Brier | $0.163_{\pm 0.016}$ | $0.160_{\pm 0.015}$ | $0.159_{\pm 0.015}$ | $0.157_{\pm 0.018}$ | $0.160_{\pm 0.015}$ | $0.159_{\pm 0.015}$ |
| NQ | AUROC | $0.659_{\pm 0.063}$ | $0.686_{\pm 0.060}$ | $0.699_{\pm 0.057}$ | $0.703_{\pm 0.060}$ | $0.702_{\pm 0.058}$ | $0.705_{\pm 0.059}$ |
| | PRR | $0.709_{\pm 0.068}$ | $0.730_{\pm 0.071}$ | $0.760_{\pm 0.065}$ | $0.764_{\pm 0.064}$ | $0.764_{\pm 0.063}$ | $0.766_{\pm 0.064}$ |
| | Brier | $0.199_{\pm 0.019}$ | $0.194_{\pm 0.020}$ | $0.184_{\pm 0.019}$ | $0.182_{\pm 0.020}$ | $0.183_{\pm 0.020}$ | $0.182_{\pm 0.021}$ |
| MMLU | AUROC | $0.604_{\pm 0.059}$ | $0.762_{\pm 0.042}$ | $0.777_{\pm 0.042}$ | $0.780_{\pm 0.041}$ | $0.790_{\pm 0.041}$ | $0.789_{\pm 0.040}$ |
| | PRR | $0.743_{\pm 0.064}$ | $0.863_{\pm 0.046}$ | $0.897_{\pm 0.031}$ | $0.898_{\pm 0.030}$ | $0.905_{\pm 0.028}$ | $0.902_{\pm 0.030}$ |
| | Brier | $0.188_{\pm 0.021}$ | $0.152_{\pm 0.017}$ | $0.148_{\pm 0.017}$ | $0.147_{\pm 0.017}$ | $0.142_{\pm 0.017}$ | $0.143_{\pm 0.017}$ |
| GSM8K | AUROC | $0.674_{\pm 0.054}$ | $0.684_{\pm 0.056}$ | $0.685_{\pm 0.054}$ | $0.695_{\pm 0.051}$ | $0.690_{\pm 0.055}$ | $0.695_{\pm 0.055}$ |
| | PRR | $0.487_{\pm 0.090}$ | $0.530_{\pm 0.096}$ | $0.527_{\pm 0.095}$ | $0.521_{\pm 0.094}$ | $0.520_{\pm 0.094}$ | $0.527_{\pm 0.095}$ |
| | Brier | $0.159_{\pm 0.021}$ | $0.151_{\pm 0.022}$ | $0.150_{\pm 0.021}$ | $0.152_{\pm 0.020}$ | $0.152_{\pm 0.020}$ | $0.146_{\pm 0.021}$ |

Table 8 shows the variation of several proposed UQ measures (Inv-Entropy, NI-Entropy, WD-px-py, MAX-py-x) as the number of replications $r$ increases. Performance improves consistently with larger $r$, as additional replications help better capture aleatoric uncertainty. However, this improvement comes at the expense of increased computational cost.

Table 8: Comparison of UQ methods on the TriviaQA and SciQ datasets with varying numbers of replications $r$. We use GPT-3.5-Turbo with ChatGPT-based paraphrasing and SBERT-large (all-mpnet-base-v2) embedding function.

| Dataset | | TriviaQA | | | SciQ | | |
|---|---|---|---|---|---|---|---|
| | Metric | $r=2$ | $r=4$ | $r=6$ | $r=2$ | $r=4$ | $r=6$ |
| Inv-Entropy | AUROC | $0.812_{\pm 0.044}$ | $0.815_{\pm 0.044}$ | $0.820_{\pm 0.044}$ | $0.794_{\pm 0.044}$ | $0.801_{\pm 0.043}$ | $0.806_{\pm 0.042}$ |
| | PRR | $0.916_{\pm 0.028}$ | $0.915_{\pm 0.029}$ | $0.915_{\pm 0.029}$ | $0.888_{\pm 0.038}$ | $0.888_{\pm 0.039}$ | $0.892_{\pm 0.038}$ |
| | Brier | $0.128_{\pm 0.020}$ | $0.121_{\pm 0.020}$ | $0.123_{\pm 0.020}$ | $0.143_{\pm 0.019}$ | $0.140_{\pm 0.019}$ | $0.136_{\pm 0.019}$ |
| NI-Entropy | AUROC | $0.702_{\pm 0.072}$ | $0.631_{\pm 0.081}$ | $0.617_{\pm 0.083}$ | $0.729_{\pm 0.056}$ | $0.731_{\pm 0.058}$ | $0.746_{\pm 0.055}$ |
| | PRR | $0.813_{\pm 0.054}$ | $0.765_{\pm 0.060}$ | $0.751_{\pm 0.059}$ | $0.805_{\pm 0.055}$ | $0.805_{\pm 0.056}$ | $0.809_{\pm 0.054}$ |
| | Brier | $0.133_{\pm 0.024}$ | $0.152_{\pm 0.023}$ | $0.150_{\pm 0.024}$ | $0.147_{\pm 0.021}$ | $0.146_{\pm 0.021}$ | $0.144_{\pm 0.021}$ |
| WD-px-py | AUROC | $0.762_{\pm 0.058}$ | $0.771_{\pm 0.058}$ | $0.772_{\pm 0.057}$ | $0.684_{\pm 0.052}$ | $0.688_{\pm 0.058}$ | $0.685_{\pm 0.057}$ |
| | PRR | $0.866_{\pm 0.046}$ | $0.875_{\pm 0.044}$ | $0.876_{\pm 0.044}$ | $0.825_{\pm 0.050}$ | $0.818_{\pm 0.054}$ | $0.819_{\pm 0.051}$ |
| | Brier | $0.134_{\pm 0.021}$ | $0.131_{\pm 0.021}$ | $0.130_{\pm 0.021}$ | $0.173_{\pm 0.017}$ | $0.169_{\pm 0.020}$ | $0.172_{\pm 0.019}$ |
| MAX-py-x | AUROC | $0.782_{\pm 0.047}$ | $0.794_{\pm 0.045}$ | $0.804_{\pm 0.044}$ | $0.754_{\pm 0.049}$ | $0.760_{\pm 0.051}$ | $0.757_{\pm 0.049}$ |
| | PRR | $0.904_{\pm 0.028}$ | $0.910_{\pm 0.027}$ | $0.915_{\pm 0.026}$ | $0.864_{\pm 0.047}$ | $0.862_{\pm 0.049}$ | $0.861_{\pm 0.048}$ |
| | Brier | $0.134_{\pm 0.019}$ | $0.130_{\pm 0.019}$ | $0.124_{\pm 0.019}$ | $0.154_{\pm 0.018}$ | $0.155_{\pm 0.019}$ | $0.154_{\pm 0.017}$ |

Table 9 demonstrates the robustness of our framework across different embedding functions. On all three datasets (TriviaQA, SciQ, and MMLU), Inv-Entropy performs strongly with SBERT-small, SBERT-large, and DeBERTa. This consistency allows practitioners to select an encoder based on available computational resources or domain-specific requirements without compromising the quality of uncertainty estimation.

Table 9: Comparison of AUROC, PRR, and Brier scores across the TriviaQA, SciQ, and MMLU datasets using different embedding functions. We use GPT-3.5-Turbo and ChatGPT-based paraphrasing.

| Dataset | Metric | Embedding | Inv-Entropy | NI-Entropy | WD-px-py | MAX-py-x |
|---------|--------|-----------|-------------|------------|----------|----------|
| **TriviaQA** | AUROC | SBERT-small | $0.792_{\pm 0.051}$ | $0.666_{\pm 0.072}$ | $0.833_{\pm 0.050}$ | $0.835_{\pm 0.046}$ |
| | | SBERT-large | $0.772_{\pm 0.057}$ | $0.617_{\pm 0.083}$ | $0.804_{\pm 0.044}$ | $0.816_{\pm 0.044}$ |
| | | DeBERTa | $0.788_{\pm 0.054}$ | $0.786_{\pm 0.057}$ | $0.518_{\pm 0.060}$ | $0.723_{\pm 0.054}$ |
| | PRR | SBERT-small | $0.899_{\pm 0.037}$ | $0.803_{\pm 0.053}$ | $0.920_{\pm 0.029}$ | $0.920_{\pm 0.029}$ |
| | | SBERT-large | $0.876_{\pm 0.044}$ | $0.751_{\pm 0.059}$ | $0.915_{\pm 0.026}$ | $0.915_{\pm 0.030}$ |
| | | DeBERTa | $0.885_{\pm 0.044}$ | $0.883_{\pm 0.043}$ | $0.763_{\pm 0.051}$ | $0.875_{\pm 0.038}$ |
| | Brier | SBERT-small | $0.127_{\pm 0.021}$ | $0.151_{\pm 0.022}$ | $0.106_{\pm 0.021}$ | $0.114_{\pm 0.021}$ |
| | | SBERT-large | $0.130_{\pm 0.021}$ | $0.150_{\pm 0.024}$ | $0.124_{\pm 0.019}$ | $0.124_{\pm 0.020}$ |
| | | DeBERTa | $0.128_{\pm 0.020}$ | $0.124_{\pm 0.020}$ | $0.184_{\pm 0.019}$ | $0.148_{\pm 0.019}$ |
| **SciQ** | AUROC | SBERT-small | $0.774_{\pm 0.049}$ | $0.750_{\pm 0.054}$ | $0.655_{\pm 0.058}$ | $0.742_{\pm 0.050}$ |
| | | SBERT-large | $0.806_{\pm 0.042}$ | $0.746_{\pm 0.055}$ | $0.685_{\pm 0.057}$ | $0.757_{\pm 0.049}$ |
| | | DeBERTa | $0.740_{\pm 0.050}$ | $0.681_{\pm 0.056}$ | $0.303_{\pm 0.060}$ | $0.674_{\pm 0.054}$ |
| | PRR | SBERT-small | $0.874_{\pm 0.044}$ | $0.820_{\pm 0.059}$ | $0.796_{\pm 0.054}$ | $0.852_{\pm 0.049}$ |
| | | SBERT-large | $0.892_{\pm 0.038}$ | $0.809_{\pm 0.054}$ | $0.819_{\pm 0.051}$ | $0.861_{\pm 0.048}$ |
| | | DeBERTa | $0.843_{\pm 0.042}$ | $0.781_{\pm 0.053}$ | $0.587_{\pm 0.056}$ | $0.821_{\pm 0.048}$ |
| | Brier | SBERT-small | $0.147_{\pm 0.021}$ | $0.150_{\pm 0.021}$ | $0.181_{\pm 0.019}$ | $0.160_{\pm 0.018}$ |
| | | SBERT-large | $0.136_{\pm 0.019}$ | $0.144_{\pm 0.021}$ | $0.172_{\pm 0.019}$ | $0.154_{\pm 0.017}$ |
| | | DeBERTa | $0.157_{\pm 0.018}$ | $0.164_{\pm 0.017}$ | $0.212_{\pm 0.016}$ | $0.177_{\pm 0.017}$ |
| **MMLU** | AUROC | SBERT-small | $0.689_{\pm 0.060}$ | $0.576_{\pm 0.063}$ | $0.723_{\pm 0.049}$ | $0.704_{\pm 0.056}$ |
| | | SBERT-large | $0.634_{\pm 0.058}$ | $0.532_{\pm 0.057}$ | $0.670_{\pm 0.055}$ | $0.676_{\pm 0.057}$ |
| | | DeBERTa | $0.780_{\pm 0.041}$ | $0.710_{\pm 0.052}$ | $0.573_{\pm 0.061}$ | $0.585_{\pm 0.059}$ |
| | PRR | SBERT-small | $0.812_{\pm 0.064}$ | $0.719_{\pm 0.069}$ | $0.869_{\pm 0.035}$ | $0.832_{\pm 0.057}$ |
| | | SBERT-large | $0.790_{\pm 0.059}$ | $0.695_{\pm 0.068}$ | $0.834_{\pm 0.049}$ | $0.820_{\pm 0.056}$ |
| | | DeBERTa | $0.898_{\pm 0.030}$ | $0.823_{\pm 0.055}$ | $0.777_{\pm 0.054}$ | $0.749_{\pm 0.062}$ |
| | Brier | SBERT-small | $0.170_{\pm 0.019}$ | $0.192_{\pm 0.020}$ | $0.163_{\pm 0.018}$ | $0.165_{\pm 0.019}$ |
| | | SBERT-large | $0.185_{\pm 0.020}$ | $0.200_{\pm 0.020}$ | $0.177_{\pm 0.019}$ | $0.173_{\pm 0.020}$ |
| | | DeBERTa | $0.147_{\pm 0.017}$ | $0.168_{\pm 0.021}$ | $0.188_{\pm 0.018}$ | $0.189_{\pm 0.020}$ |

Table 10 and Table 11 present how various uncertainty measures respond to temperature changes on the TriviaQA and SciQ datasets, respectively. Our probabilistic methods consistently outperform baseline models across all temperature settings, highlighting their robustness to decoding variability.

Table 10: Comparison of AUROC, PRR, and Brier scores on the TriviaQA dataset under varying temperatures ($\mathbb{T} = 0.3,\ 0.7,\ 1.0,\ 1.4$). We use GPT-3.5-Turbo with ChatGPT-based paraphrasing and the SBERT-small (paraphrase-MiniLM-L6-v2) embedding function. **Bold** and underline indicate the best and second-best performers, respectively.

| Metric | Method | Temperature | | | |
|---|---|---|---|---|---|
| | | $\mathbb{T}=0.3$ | $\mathbb{T}=0.7$ | $\mathbb{T}=1.0$ | $\mathbb{T}=1.4$ |
| AUROC($\uparrow$) | Semantic Entropy | $0.579_{\pm 0.044}$ | $0.679_{\pm 0.052}$ | $0.700_{\pm 0.062}$ | $0.732_{\pm 0.051}$ |
| | VU | $0.695_{\pm 0.060}$ | $0.625_{\pm 0.058}$ | $0.629_{\pm 0.063}$ | $0.578_{\pm 0.067}$ |
| | P(True) | $0.604_{\pm 0.050}$ | $0.592_{\pm 0.046}$ | $0.571_{\pm 0.043}$ | $0.583_{\pm 0.038}$ |
| | LexSim | $0.649_{\pm 0.055}$ | $0.715_{\pm 0.051}$ | $0.775_{\pm 0.055}$ | $0.792_{\pm 0.051}$ |
| | DegMat | $0.734_{\pm 0.056}$ | $0.688_{\pm 0.067}$ | $0.794_{\pm 0.049}$ | $0.730_{\pm 0.059}$ |
| | LUQ | $0.637_{\pm 0.067}$ | $0.712_{\pm 0.059}$ | $0.748_{\pm 0.063}$ | $0.817_{\pm 0.044}$ |
| | KLE | $0.333_{\pm 0.054}$ | $0.327_{\pm 0.059}$ | $0.221_{\pm 0.056}$ | $0.155_{\pm 0.042}$ |
| | Inv-Entropy | $\mathbf{0.870}_{\pm 0.044}$ | $0.795_{\pm 0.051}$ | $\underline{0.827}_{\pm 0.043}$ | $0.810_{\pm 0.042}$ |
| | NI-Entropy | $0.762_{\pm 0.063}$ | $0.666_{\pm 0.073}$ | $0.634_{\pm 0.083}$ | $0.673_{\pm 0.055}$ |
| | WD-px-py | $0.829_{\pm 0.044}$ | $\underline{0.827}_{\pm 0.051}$ | $0.816_{\pm 0.054}$ | $\underline{0.829}_{\pm 0.039}$ |
| | MAX-py-x | $\underline{0.854}_{\pm 0.043}$ | $\mathbf{0.832}_{\pm 0.046}$ | $\mathbf{0.856}_{\pm 0.040}$ | $\mathbf{0.846}_{\pm 0.041}$ |
| PRR($\uparrow$) | Semantic Entropy | $0.787_{\pm 0.044}$ | $0.803_{\pm 0.046}$ | $0.834_{\pm 0.043}$ | $0.811_{\pm 0.045}$ |
| | VU | $0.836_{\pm 0.044}$ | $0.791_{\pm 0.041}$ | $0.798_{\pm 0.053}$ | $0.727_{\pm 0.051}$ |
| | P(True) | $0.797_{\pm 0.042}$ | $0.760_{\pm 0.044}$ | $0.776_{\pm 0.042}$ | $0.726_{\pm 0.046}$ |
| | LexSim | $0.810_{\pm 0.045}$ | $0.824_{\pm 0.040}$ | $0.877_{\pm 0.043}$ | $0.854_{\pm 0.043}$ |
| | DegMat | $0.882_{\pm 0.041}$ | $0.816_{\pm 0.053}$ | $0.895_{\pm 0.037}$ | $0.812_{\pm 0.054}$ |
| | LUQ | $0.854_{\pm 0.043}$ | $0.856_{\pm 0.042}$ | $0.874_{\pm 0.048}$ | $0.893_{\pm 0.039}$ |
| | KLE | $0.704_{\pm 0.048}$ | $0.646_{\pm 0.050}$ | $0.623_{\pm 0.051}$ | $0.516_{\pm 0.049}$ |
| | Inv-Entropy | $\mathbf{0.939}_{\pm 0.028}$ | $0.900_{\pm 0.037}$ | $\underline{0.936}_{\pm 0.022}$ | $0.903_{\pm 0.037}$ |
| | NI-Entropy | $0.862_{\pm 0.043}$ | $0.799_{\pm 0.054}$ | $0.777_{\pm 0.060}$ | $0.765_{\pm 0.055}$ |
| | WD-px-py | $\underline{0.938}_{\pm 0.021}$ | $\underline{0.918}_{\pm 0.029}$ | $0.912_{\pm 0.043}$ | $\mathbf{0.923}_{\pm 0.025}$ |
| | MAX-py-x | $0.932_{\pm 0.029}$ | $\mathbf{0.920}_{\pm 0.030}$ | $\mathbf{0.946}_{\pm 0.021}$ | $\underline{0.913}_{\pm 0.039}$ |
| Brier($\downarrow$) | Semantic Entropy | $0.166_{\pm 0.023}$ | $0.150_{\pm 0.026}$ | $0.141_{\pm 0.023}$ | $0.156_{\pm 0.023}$ |
| | VU | $0.160_{\pm 0.022}$ | $0.178_{\pm 0.017}$ | $0.175_{\pm 0.023}$ | $0.203_{\pm 0.016}$ |
| | P(True) | $0.172_{\pm 0.022}$ | $0.188_{\pm 0.020}$ | $0.179_{\pm 0.021}$ | $0.198_{\pm 0.020}$ |
| | LexSim | $0.151_{\pm 0.024}$ | $0.146_{\pm 0.021}$ | $0.128_{\pm 0.024}$ | $0.127_{\pm 0.021}$ |
| | DegMat | $0.140_{\pm 0.021}$ | $0.149_{\pm 0.022}$ | $\underline{0.115}_{\pm 0.018}$ | $0.145_{\pm 0.023}$ |
| | LUQ | $0.148_{\pm 0.020}$ | $0.142_{\pm 0.021}$ | $0.121_{\pm 0.023}$ | $\underline{0.121}_{\pm 0.018}$ |
| | KLE | $0.188_{\pm 0.021}$ | $0.199_{\pm 0.020}$ | $0.192_{\pm 0.021}$ | $0.218_{\pm 0.014}$ |
| | Inv-Entropy | $\mathbf{0.085}_{\pm 0.019}$ | $0.127_{\pm 0.021}$ | $0.117_{\pm 0.019}$ | $0.132_{\pm 0.017}$ |
| | NI-Entropy | $0.104_{\pm 0.021}$ | $0.151_{\pm 0.022}$ | $0.142_{\pm 0.024}$ | $0.164_{\pm 0.018}$ |
| | WD-px-py | $0.115_{\pm 0.018}$ | $\mathbf{0.102}_{\pm 0.022}$ | $0.117_{\pm 0.021}$ | $\underline{0.121}_{\pm 0.017}$ |
| | MAX-py-x | $\underline{0.103}_{\pm 0.019}$ | $\underline{0.116}_{\pm 0.021}$ | $\mathbf{0.108}_{\pm 0.019}$ | $\mathbf{0.114}_{\pm 0.016}$ |

Table 11: Comparison of AUROC, PRR, and Brier scores on the SciQ dataset under varying temperatures ($\mathbb{T} = 0.3$, 0.7, 1.0, 1.4). We use GPT-3.5-Turbo with ChatGPT-based paraphrasing and the SBERT-large (all-mpnet-base-v2) embedding function. **Bold** and underline indicate the best and second-best performers, respectively.

| Metric | Method | Temperature | | | |
|---|---|---|---|---|---|
| | | $\mathbb{T}$=0.3 | $\mathbb{T}$=0.7 | $\mathbb{T}$=1.0 | $\mathbb{T}$=1.4 |
| AUROC(↑) | Semantic Entropy | $0.548_{\pm0.040}$ | $0.679_{\pm0.045}$ | $0.699_{\pm0.046}$ | $0.773_{\pm0.043}$ |
| | VU | $0.625_{\pm0.061}$ | $0.480_{\pm0.060}$ | $0.625_{\pm0.050}$ | $0.607_{\pm0.065}$ |
| | P(True) | $0.566_{\pm0.037}$ | $0.522_{\pm0.026}$ | $0.531_{\pm0.026}$ | $0.545_{\pm0.026}$ |
| | LexSim | $0.552_{\pm0.041}$ | $0.681_{\pm0.046}$ | $0.713_{\pm0.049}$ | $0.770_{\pm0.051}$ |
| | DegMat | $0.569_{\pm0.065}$ | $0.672_{\pm0.059}$ | $0.730_{\pm0.055}$ | $\mathbf{0.816}_{\pm0.040}$ |
| | LUQ | $0.565_{\pm0.062}$ | $0.740_{\pm0.050}$ | $0.650_{\pm0.060}$ | $0.772_{\pm0.051}$ |
| | KLE | $0.386_{\pm0.054}$ | $0.341_{\pm0.056}$ | $0.277_{\pm0.059}$ | $0.184_{\pm0.051}$ |
| | Inv-Entropy | $\mathbf{0.767}_{\pm0.047}$ | $\mathbf{0.800}_{\pm0.044}$ | $0.754_{\pm0.046}$ | $0.795_{\pm0.046}$ |
| | NI-Entropy | $0.682_{\pm0.062}$ | $0.755_{\pm0.055}$ | $0.596_{\pm0.065}$ | $0.649_{\pm0.063}$ |
| | WD-px-py | $0.711_{\pm0.053}$ | $0.689_{\pm0.056}$ | $0.795_{\pm0.047}$ | $0.771_{\pm0.045}$ |
| | MAX-py-x | $0.766_{\pm0.048}$ | $0.760_{\pm0.048}$ | $\mathbf{0.796}_{\pm0.046}$ | $0.793_{\pm0.044}$ |
| PRR(↑) | Semantic Entropy | $0.705_{\pm0.045}$ | $0.763_{\pm0.044}$ | $0.780_{\pm0.043}$ | $0.798_{\pm0.046}$ |
| | VU | $0.739_{\pm0.062}$ | $0.677_{\pm0.053}$ | $0.736_{\pm0.053}$ | $0.678_{\pm0.061}$ |
| | P(True) | $0.713_{\pm0.044}$ | $0.679_{\pm0.050}$ | $0.687_{\pm0.045}$ | $0.634_{\pm0.045}$ |
| | LexSim | $0.701_{\pm0.052}$ | $0.770_{\pm0.051}$ | $0.794_{\pm0.047}$ | $0.796_{\pm0.056}$ |
| | DegMat | $0.739_{\pm0.060}$ | $0.802_{\pm0.046}$ | $0.834_{\pm0.046}$ | $0.864_{\pm0.040}$ |
| | LUQ | $0.709_{\pm0.058}$ | $0.843_{\pm0.042}$ | $0.734_{\pm0.061}$ | $0.813_{\pm0.061}$ |
| | KLE | $0.632_{\pm0.048}$ | $0.592_{\pm0.059}$ | $0.554_{\pm0.052}$ | $0.479_{\pm0.051}$ |
| | Inv-Entropy | $\mathbf{0.900}_{\pm0.029}$ | $\mathbf{0.888}_{\pm0.039}$ | $0.864_{\pm0.044}$ | $\mathbf{0.870}_{\pm0.035}$ |
| | NI-Entropy | $0.814_{\pm0.047}$ | $0.820_{\pm0.055}$ | $0.707_{\pm0.059}$ | $0.693_{\pm0.064}$ |
| | WD-px-py | $0.858_{\pm0.040}$ | $0.820_{\pm0.052}$ | $0.868_{\pm0.050}$ | $0.851_{\pm0.037}$ |
| | MAX-py-x | $0.893_{\pm0.029}$ | $0.862_{\pm0.048}$ | $\mathbf{0.887}_{\pm0.037}$ | $0.867_{\pm0.034}$ |
| Brier(↓) | Semantic Entropy | $0.203_{\pm0.018}$ | $0.173_{\pm0.020}$ | $0.171_{\pm0.021}$ | $0.159_{\pm0.022}$ |
| | VU | $0.191_{\pm0.022}$ | $0.196_{\pm0.017}$ | $0.202_{\pm0.018}$ | $0.212_{\pm0.016}$ |
| | P(True) | $0.204_{\pm0.017}$ | $0.215_{\pm0.017}$ | $0.212_{\pm0.016}$ | $0.225_{\pm0.013}$ |
| | LexSim | $0.207_{\pm0.020}$ | $0.179_{\pm0.020}$ | $0.169_{\pm0.022}$ | $0.156_{\pm0.019}$ |
| | DegMat | $0.198_{\pm0.017}$ | $0.164_{\pm0.018}$ | $0.149_{\pm0.020}$ | $\mathbf{0.135}_{\pm0.020}$ |
| | LUQ | $0.195_{\pm0.020}$ | $0.157_{\pm0.018}$ | $0.173_{\pm0.021}$ | $0.159_{\pm0.020}$ |
| | KLE | $0.216_{\pm0.016}$ | $0.218_{\pm0.016}$ | $0.219_{\pm0.016}$ | $0.233_{\pm0.011}$ |
| | Inv-Entropy | $\mathbf{0.151}_{\pm0.018}$ | $\mathbf{0.139}_{\pm0.020}$ | $0.153_{\pm0.020}$ | $0.146_{\pm0.020}$ |
| | NI-Entropy | $0.164_{\pm0.021}$ | $0.140_{\pm0.022}$ | $0.188_{\pm0.020}$ | $0.186_{\pm0.020}$ |
| | WD-px-py | $0.167_{\pm0.018}$ | $0.175_{\pm0.019}$ | $\mathbf{0.140}_{\pm0.020}$ | $0.156_{\pm0.015}$ |
| | MAX-py-x | $0.151_{\pm0.019}$ | $0.155_{\pm0.018}$ | $0.143_{\pm0.020}$ | $0.153_{\pm0.018}$ |

