# OpenReview forum: "Inv-Entropy: A Fully Probabilistic Framework for Uncertainty Quantification in Language Models"
_NeurIPS.cc/2025/Conference — NeurIPS 2025 poster_

### Official Review · Reviewer_cePH · 2025-06-16

**Clarity:** 1
**Significance:** 2
**Originality:** 3
**Rating:** 2
**Confidence:** 2

**Summary:**

The paper considers the uncertainty quantification of LLM question-answering tasks. They propose to a set of prompt variations, and perform random walks between their embeddings, and derive some entropy measures from them. I think the work is measuring how much the prompt variations change the embeddings, but I’m not totally sure. The results show excellent performance on UQ benchmarks.

**Questions:**

- I overall struggled to understand section 2. I couldn’t connect the presentation to the overall LLM setting (ie. prompt, LLM, next-tokens, responses) or the underlying problem. I also struggled to follow the presentation mathematically due to lack of rigor (eg. not defining variables, functions or their domains). It would be very useful to give a real-world LLM example and denote how all the presented variables and functions relate to it.
- What is “semantic input”? Are there other types of inputs as well? What does S_x0 mean? What does x0 mean? What are their sizes or domains? Is S a function?
- What does “LLM” f actually mean? LLMs are quite big models: is f the input-response mapping, or the model architecture, or some abstract model concept? What are their inputs and outputs?
- What does “semantic embedding” psi mean? Is this related to transformers and next-token predictions, or is this some external model? What are their inputs and outputs?
- We have x_i = psi(S(x_i)): it seems that nothing happens. Why do we do this, and what does this mean?
- Xn and Yn contain “samples”. Why are elements of Xn samples? I see nothing stochastic here. Are the samples from some distribution? If so, how is it defined? Is f stochastic? In what way?
- What does X,Y mean? Are they related to Xn,Yn or xi,yi?
- What are the two markov chains? How are they defined? Where is the sequential (time) indexing in the markov chain?
- What are the input and output spaces?
- In 140 the x0 is described as “question”, but earlier x0 was embedding. Is it both?
- Eq 3 defines a “marginal” probability that clearly is not marginal probability. Marginal probability describes chance of being in state j no matter where we are in the process without knowing past or future. Eq 3 is some kind of transition probability instead.
- I can’t understand the \pi P_y[j] notation. P_y is not defined. \pi as operator is not defined.
- Eq 4 is a conditional probability of moving from j to i through some intermediate k. I don’t understand why we do this. The notation should be explicit about how many steps there are. I don’t understand how can we mix the Px and Py matrices here, or what it means. I don’t understand why we move y from x to j, but x from k to i.
- It’s quite odd that the markov presentation does not contain temporal indexing. Markov chain is a sequence of random variables indexed by time, ie. X_t \in {0,1,…,n}. Here the time is not present, but instead it seems that X and Y are time-independent (but also a chain…), or maybe they are just some kind of 1 or 2 step chains that never proceed further [but that’s surely not a markov chain]. I find this quite confusing. Can you discuss?
- I don’t understand why line 170 introduces a 3-step probability, or why it’s called “marginal”.
- Why do you only consider few-step transition probabilities, and not true marginals of the stationary distribution of the markov chain?
- I don’t understand what conditional sample entropy means here. Entropy is only defined for random variables with known densities. Samples don’t come with densities. Furthermore, conditional entropy and entropy of a conditional are different things. I think the latter is being used here, but wording implies former. Can you clarify?
- Can you try to clarify what the entropy eq 5 measures conceptually?
- What is R_i?
- After reading the sec 2, I’m a bit confused what did we achieve? It seems that bunch of variations on some prompt are generated. We then quantify how aligned the embeddings of the prompt and its answer are, and compute an entropy of some transitions between variations. I don’t see how this now connects with the tangent space / invariance discussion of sec 2.1, or what this now measures or quantifies. If the \hat{H} is now high or low, is that good or bad? What do we want to see or achieve here? Do we want to see the the correct pairings (xi,yi) being most probable [why or why not]? What does all of this tell us about LLMs?
- What do you actually do in the experiments? Do you train something? Do you change the LLM prediction logits at test time in some way? The alg1 tells us that you compute hatH: how is the hatH used to improve the produced logits?
- What is the nature of uncertainty you quantify? What is “uncertainty” in this paper?
- The results seem to be contain methods that are not described in the paper

**Ethical Concerns:**

["NO or VERY MINOR ethics concerns only"]

**Final Justification:**

The authors have engaged in informative discussion during response period and the contributions have become clearer. I retain my reject rating due to presentational issues. The paper has some disconnect between the tangent space and random walk arguments, and the random walk presentation is not mathematically sufficiently rigorous. The method is a black-box, and there is little attempt to try to give insights behind the produced uncertainty summary metrics.

**Limitations:**

No issues.

**Paper Formatting Concerns:**

No issues

**Quality:**

2

**Strengths And Weaknesses:**

The idea of using embeddings of prompt and output variations is generally sensible, but I did not fully understand what happens in this paper, so I’m not totally sure of this. The results are excellent, which indicates the method works and does something.

The clarify of the paper is subpar. The method description is conceptually vague, and I could not concretise the presented ideas to existing LLM ideas. The method description also lacks rigor, which makes the math difficult to follow or verify. The paper requires a major revision on both counts.

I’m not totally sure if the method is well-defined or well-motivated. There is lot of sloppyness in the math, and I don’t understand why the paper considers random walks of just length 2 or 3. Surely one should look at the markov chain stationary distribution. Moreover, I could not understand why random walks are considered in the first place: couldn’t we just measure the concentration of the embeddings more directly?

Despite the great results, the paper is not yet ready for neurips and requires a major overhaul of the presentation to verify the method and its math are sound.

---

> ### Author Rebuttal · Authors · 2025-07-29
>
> Thank you for your candid feedback.
>
> >**Regarding Clarity & Mathematical Rigor**:
>
> **Clarity**: We respectfully note that the paper was consistently praised by other reviewers for its readability and accessibility. Reviewer Q6Vh described it as “well written and easy to follow,” Reviewer uG7q called it “exceptionally clearly written,” and Reviewer tjtF similarly stated that it is “well-written and easy to follow.” We appreciate that you noted difficulty in connecting the concepts to existing LLM ideas. However, we would have greatly appreciated more specific pointers to where the exposition could be improved so that we could make targeted revisions.
>
> **Mathematical rigor**: We respectfully disagree with the claim that the paper lacks mathematical rigor. Section 2.1 is built on foundational tools from functional analysis and probabilistic modeling, and all core quantities are well-defined. We thoroughly reviewed your comments and found no substantive issues in the mathematics. In fact, other reviewers highlighted the strength of our theoretical contribution; for example, Reviewer Q6Vh emphasized the “highly valuable theoretical perspective,” and Reviewer uG7q described the method as “elegant,” awarding the paper top marks for both quality and originality. If there are particular definitions or results you believe are incorrect or unclear, we would be eager to clarify them.
>
> We sincerely appreciate your questions and respond to each of your 23 points in detail below, and hope our clarifications help improve clarity and reinforce our contributions.
>
> >**Random walk**
>
> *Why do we consider random walk*: In its core, Inv-entropy examines the alignment between LLM inputs and output embedding. Random walk is a tool we introduce to calculate such alignments. Similarity-based random walk is extremely popular in structure identification [1] and data alignment [2]. Since it is intuitive and flexible, we design our uncertainty measure with the help from random walk.
>
> *Why do we consider random walk with short steps*: Indeed, there are multiple ways to characterize the similarity between the two transition probability. In our paper, we examined the performance of $5$ metrics, Inv-Entropy, NI-Entropy, NR-Inv-Entropy, WD-px-py, MAX-py.  Table 1 shows that Inv-Entropy yields the best performance.
>
> Based on your suggestion, we replace $P(Y)$ in (3) by the stationary distribution of $\text{P}_y$ and replace the conditional distribution $P(X_i|Y_i)$ by the stationary distribution of $\text{P}_y\text{P}_x$. The comparison in performance of Inv-entropy in the TriviaQA dataset (other settings are the same as in the main paper) is provided in the table below.
>
> | Metric         | Stationary | Non-Stationary  |
> |----------------|-------|-------|
> | PRR            |    0.697 ± 0.043    |    0.902 ± 0.042      |
> | Brier          |  0.209 ± 0.017    |    0.117 ± 0.020     |
>
> The results highlight that using the stationary distribution yields much worse performance. This outcome is further explained below.
>
> [Q15,Q17] We only use two steps of random walk in Inv-Entropy is optimal for two reasons to introduce an implicit bias. The defined marginal distribution $P(Y)$ is the marginal distribution of $n+1$ states after one step of random walk from uniform distribution under $\text{P}_y$. This provides an implicit bias over $P(Y)$ that it should not differ too much from uniform distribution $\pi^{\text{Uniform}}$. Numerically, such an inductive bias provides stability in our estimate. If we instead use the stationary distribution of $\text{P}_y$ as $P(Y)$, the distribution could significantly deviate from the initial $\pi^{\text{Uniform}}$, which results in worse performance in practice.
>
> [Q8,Q14] Since we only consider the Markov chains that mix for at most two steps, we do not introduce a time subscript to simplify notations.
>
> >**The rigor of Section 2.1**
>
> [Q1] Section 2.1 is based on rigorous functional analysis. The domain of $f$ is $\mathbb{R}^d$ and the output is a scalar in $\mathbb{R}$. The definitions are elaborated in line $74$ to $87$. We are glad to clarify any potential misunderstanding.
>
> We also acknowledge that the rigorous analysis in section 2.1 is only a simplification of realistic LLMs, whose inputs are token sequences and outputs are also probabilities in the token space. Nevertheless, we can use $f$ to denote the mapping from LLM input token sequences to one embedding coordinate of LLM output tokens. The purpose of Lemma 2.1 is explained in line 97 to 99.
>
> >**Notations about tokens and embeddings**
>
> [Q2,Q3] Semantic input $S_{x_0}$ is input token sequences to LLM. It is defined in line 105.
>
> [Q4] Input embedding $x_0$ is the embedding of the input sequence $S_{x_0}$ extracted by an external model. We intentionally leave the choice of this external model unspecified to provide practitioners with flexibility in implementation.
>
> [Q5] $\psi(S_{x_i})$ is the process of converting token sequences to their embedding.
>
> [Q6] $X_n$ is a set of input sequences and $Y_n$ is the set of corresponding LLM outputs. A more appropriate name should be “elements”.
>
> [Q9] The input space and output space are $X_n$ and $Y_n$, respectively.
>
> >**Definitions about marginal and conditional probability**
>
> [Q7,Q11] In line 130, $X$ and $Y$ are defined as discrete random variables whose supports are $X_n$ and $Y_n$. In principle, there are infinitely multiple ways to define the marginal and conditional probability of $X$ and $Y$. This notation of marginal or conditional should not be confused with the dynamics related to Markov chains. [Q16] This is also the case for line 170.
>
> [Q8,Q12] We only use Markov chains as probabilistic tools to construct the marginal of $p(Y)$ and the conditional distribution $p(X|Y)$. The states, indices, and transition probabilities of the two Markov chains are defined from line 116 to line 123. $\text{P}_x$ and $\text{P}_y$ are two matrices defined in (2). $\pi^{\text{Uniform}}$ is a vector defined in line 136 to 137.
>
> [Q13] In (4), the summation form of $P(X=x_i|Y=y_j)$ is the direct application of matrix multiplication of the two matrices defined in (2).
>
> [Q10] In line 140, the original question should be $S_{x_0}$, rather than $x_0$.
>
> [Q16] The three step transition defined in line 170 is a natural application of the Bayes rule based on (3) and (4).
>
> >**Motivation for (5)**
>
> [Q17] Thank you for pointing out your concerns about our naming. In fact, (5) is neither entropy or conditional entropy in the standard sense. It is the Shannon entropy of the diagonal entries of $\text{P}_y\text{P}_x$, which is dependent on $X_n$ and $Y_n$. Since $X_n$ consists of samples from the input perturbation algorithm and $Y_n$ contains their corresponding outputs, we call (5) conditional sample entropy.
>
> [Q18] For a more intuitive explanation, suppose $\zeta_0,\zeta_2,\cdots,\zeta_{n}$ are the diagonal entries of matrix $\text{P}_y\text{P}_x$. Then $\zeta_i$ measures how likely is it that, starting at point $i$, a two-step walk through $\text{P}_y$ then through $\text{P}_x$ will return to $i$.
>
> As a result, $H(X_n|Y_n)$ in (5) is equal to $H(X_n|Y_n)=\sum_{i=0}^{n} - \zeta_i\log\zeta_i$, which decreases with larger $\zeta_i$. It measures the uncertainty or diversity in self-transition behavior across all states under the combined dynamics of $\text{P}_y\text{P}_x$. Low entropy often correspond to high $s_i$'s, which indicate $\text{P}_y$ and $\text{P}_x$ are more aligned, which is considered to be good.
>
> [Q21] Section 2.1 introduces a simplified theoretical prototype to justify the perturb-then-estimate foundation of Inv-Entropy. The tangent plane is a mathematical abstraction of the class of semantic invariant inputs to LLMs. Section 2.1 says if $f(\cdot)$ generates very different outputs to inputs that should have similar outputs, then $f$ is probably inaccurate, which is bad.
>
> However, in practice, generating diverse yet exactly semantic equivalent input variants is almost impossible. That is why Inv-Entropy also introduces an input similarity graph and uses a random walk on such graphs to characterize small input semantic variations. Our defined $H$ is a natural statistics that describes the local alignment between input and outputs. This is the nature of our measure of uncertainty.
>
> [Q19] $R_i$ is the set of all LLM response by asking LLM the same question $S_{x_i}$ multiple times.
>
>
> >**Clarifications about our numerical results**
>
> [Q22] In our experiments, we did not train nor improve the logits of LLMs: that's not the task of Inv-Entropy. Instead, we take existing LLMs and evaluate their uncertainty. $\hat{H}$ is a metric of how uncertain the LLM output is. We cross-validate $\hat{H}$ with other established measures of uncertainty, including correctness or uncertainty distribution, using AUROC, PRR, and Brier score. Our experiment setup is explained from line 242 to line 252.
>
> [Q23] All methods are defined from line 233 to 241.
>
> **Reference**
>
> [1] Belkin, Mikhail, and Partha Niyogi. "Laplacian eigenmaps for dimensionality reduction and data representation." Neural computation 15.6 (2003): 1373-1396.
>
> [2] Talmon, Ronen, and Hau-Tieng Wu. "Latent common manifold learning with alternating diffusion: analysis and applications." Applied and Computational Harmonic Analysis 47.3 (2019): 848-892.

---

> > ### Comment · Reviewer_cePH · 2025-08-04
> >
> > Thanks for the response, many of my confusions have been clarified.
> >
> > My main concerns with the paper are in the methodological motivation and clarity. The idea of measuring the input/output alignment over perturbations is a sensible one, but I find the paper’s approach to a bit adhoc.
> >
> > Sec 2 motivates the work by discussing tangent spaces and Taylor expansions. I like this approach, but I don’t see how this connects with the later probabilistic approach. There seems to be a confusing disconnect .
> >
> > The main innovation of the paper is studying the 2-step path probabilities of the perturbed input/output embeddings. I can see that this can be practically useful, and it measures some kind of isolated'ness of the embeddings, but I don’t see a clear motivation or what the method precisely measures. Furthermore, there is also no empirical illustration of the method either: We don’t see what the Xn/Yn/Py/Px look like in practise so the method is black-box. I’m not sure if we gain much insight about LLM perturbation or uncertainty behavior from this method, apart from the uncertainty numbers in the tables.
> >
> > Finally, I find issues in the eq 5. Entropy is a measure of a distribution, but eq 5 LHS defines an entropy of a set of samples [samples are not random variables or distributions]. The paper also defines Xn as the “support” of X, but X is also just a set of samples. I’m not sure I can follow what’s going on [maybe Xn is interpreted as a mixture of Dirac distribution?]. The eq 5 RHS looks like conditional entropy, but it sums over samples, while it should sum over the domain instead. I’m also not sure what is “inverse” about eq 5. This part overall lacks rigor. In the response this is described as diagonal entropy. That requires one to establish that the diagonal forms a valid distribution, which is not shown in the paper. I'm not sure if entropy is the correct concept for studying how concentrated the transition matrix diagonal is.
> >
> > Overall the good results clearly indicate method’s potential, but the motivation and mathematical presentation are both a bit too “loose”, and I don’t think this is yet ready for publication.

---

> ### Author Response · Authors · 2025-08-05
> **Follow-up**
>
> Dear Reviewer cePH,
>
> We are glad to hear that many of your confusions have been clarified. Please allow us to address your remaining concerns:
>
> >**Main Concern**
>
> We acknowledge that all existing UQ methods for LLMs are currently ad hoc, given the complexities of natural language and transformer architectures. Indeed, this is a field still in its infancy, and much remains to be explored. While we do not claim to have solved the explainability problem, we believe our work represents a significant step forward in quantifying LLM uncertainty inspired by a probabilistic framework. The strong empirical performance across all datasets and evaluation metrics highlights the potential of our approach.
>
> >**Gaining Insights**
>
> As emphasized in both the literature and our paper, UQ in language cannot readily provide interpretable summaries like confidence intervals for continuous variables. Instead, the goal of this growing body of work is to derive metrics that are highly correlated with the LLM's underlying knowledge, so they can inform downstream decisions: such as when to stop a reasoning chain or flag potential hallucinations.
>
> >**Theory**
>
> While the initial motivation uses tangent spaces and first-order Taylor expansions, it serves to build geometric intuition for why a **perturb-then-estimate framework** should relate to semantic shifts in the output space. The probabilistic approach that follows formalizes this framework by modeling transitions between these perturbed samples using Markov chains, offering practitioners an effective tool to analyze how well LLM outputs align with input variations.
>
>
> >**Clarifying Some Misunderstandings**
>
> Throughout the paper, we define the capital letter $X$ as a discrete random variable whose values lie in the finite set $X_n$. Similarly, $Y$ is a random variable with values in $Y_n$. In your language, the distributions of $X$ and $Y$ are the mixture of Dirac distributions, and we are only defining the mixture coefficients by (3) and (4). These definitions are introduced to motivate (5). **Could you please further clarify what mathematical or notational inconsistencies you found ?**
>
> The probabilistic framework in Sections 2.2 and 2.3 aims to justify the construction of (5) using $X_n$ and $Y_n$. We interpret $\text{P}_x$ and $\text{P}_y$ (defined in (2)) as transition probabilities of two Markov chains, where the states correspond to elements of $X_n$ and $Y_n$, respectively. Then, Eq. (5) naturally corresponds to the “entropy” of self-transition probabilities under the composite dynamics of $\text{P}_x$ and $\text{P}_y$.
>
> Importantly, what we refer to as conditional sample entropy is not the entropy of a probability distribution. We use the term “conditional sample entropy” only because its form matches that of Shannon entropy. In fact, it is an unnormalized entropy, **intentionally designed** to serve as our uncertainty measure.
>
> In the initial rebuttal, we define $\zeta_i$ as a diagonal element of $\text{P}_y\text{P}_x$, where $\zeta_i$ corresponds to $p(x_i \mid y_i)$ as defined in our main paper. Let $\Xi = \sum_i \zeta_i$. We can normalize as:
>
> $$
> \tilde{\zeta}_i = \frac{\zeta_i}{\Xi}.
> $$
>
> $\tilde{\boldsymbol{\zeta}} = (\tilde{\zeta}_1, \ldots, \tilde{\zeta}_n) $ then forms a valid probability distribution over $X_n$.
>
> The normalized Shannon entropy is defined as:
>
> $$
> \text{Entropy}(X_n \mid Y_n) = -\sum_i \tilde{\zeta}_i\log \tilde{\zeta}_i
> $$
>
> Hence, our Inv-Entropy becomes:
>
> $$
> H(X_n \mid Y_n) = -\sum_i \zeta_i \log \zeta_i = \Xi  \text{Entropy}(X_n \mid Y_n) - \Xi \log \Xi
> $$
>
> This expression aligns with our goal of modeling **two sources of uncertainty**:
>
> **(1) Total mass $ \Xi $:** $\Xi$ reflects the overall input-output specificity. Higher values indicate more isolated mappings, which we treat as a factor contributing to uncertainty
>
> **(2) Dispersion of $\tilde{\boldsymbol{\zeta}}$:** Entropy over $\tilde{\zeta}_i$ captures variability across perturbations. Uniform patterns imply stability, while peaked ones reflect inconsistency.
>
> We will revise the main paper to emphasize this interpretive lens for improved clarity.
>
> >**Inverse Perspective**
>
> Eq. (5) is referred to as "inverse" because it measures the diagonal entries of $\text{P}_y\text{P}_x$, which is dependent on the diversity of elements in $X_n$. On the contrary, ordinary entropy often focuses only on the distribution related to $\text{P}_y$. Based on the reasoning in Section 2.2, Inv-Entropy captures uncertainty through the lens of $P(X|Y)$.
> We would like to refer the reviewer to our response to **Reviewer uG7q under “W4: Extreme-Case Analysis”**, which provides a theoretical ablation for the case where perturbing the input yields the same output. This analysis demonstrates that as the diversity of the input increases, Inv-Entropy decreases;  highlighting the sensitivity of the approach and its ability to reflect the diversity of the input space that could have produced a given response.

---

> > ### Comment · Reviewer_cePH · 2025-08-05
> >
> > Thanks for the discussion, and I appreciate your efforts on clarifying your contributions.

---

> > > ### Author Response · Authors · 2025-08-05
> > > **Grade**
> > >
> > > Dear Reviewer cePH,
> > >
> > > Thank you for your engagement. We greatly appreciate it.
> > >
> > > As you currently hold the lowest score among the reviewers, we wanted to kindly ask: is there any additional clarification or further analysis we could provide that might help you reconsider your evaluation ?

---

### Official Review · Reviewer_pA2S · 2025-06-17

**Clarity:** 2
**Significance:** 4
**Originality:** 3
**Rating:** 4
**Confidence:** 4

**Summary:**

This paper presents Inv-Entropy, a novel fully probabilistic framework for uncertainty quantification (UQ) in large language models (LLMs). It introduces an inverse modeling perspective grounded in random walk theory to characterize the diversity of inputs that could yield a specific output. Key contributions include a dual Markov chain formulation over input–output semantic embeddings and a new UQ metric (Inv-Entropy), supported by Monte Carlo bootstrapping. Further proposals are a GAAP perturbation algorithm utilizing genetic operations to enhance input diversity and an evaluation metric called Temperature Sensitivity of Uncertainty (TSU) that does not rely on correctness labels. The method is evaluated across various datasets and compared to state-of-the-art uncertainty quantification (UQ) baselines, demonstrating consistent improvements.

**Questions:**

- Past literature [2,4,5] did not use a separate holdout set for their evaluation. Their approaches have likely been optimized on the test set. Did the authors follow this practice, or was a holdout set used? And if not, what is the justification?
- How are the hyperparameters chosen? Over what range of hyperparameters was optimized?
- Lemma 2.1 uses the variance of a distribution, which depends on the tangent of $f^*$. However, $f^*$ is not known in practice. How does this theory and its discussion relate to the algorithm used in the experiments?

**References (continued)**

- [4] Kuhn, L., Gal, Y., & Farquhar, S. (2023). Semantic uncertainty: Linguistic invariances for uncertainty estimation in natural language generation. ICLR.

- [5] Nikitin, A., Kossen, J., Gal, Y., & Marttinen, P. (2024). Kernel language entropy: Fine-grained uncertainty quantification for llms from semantic similarities. NeurIPS.

**Ethical Concerns:**

["NO or VERY MINOR ethics concerns only"]

**Final Justification:**

Most of my concerns have been solved, which warrants an increase in the score.
However, my major concerns (and this is shared by another reviewer) are the clarity and the presentation, which would require me to reread the revised manuscript.
This is important since the theoretical framework is introduced without citing the relevant literature it is based on, and TSU is presented as a contribution but not introduced and discussed in the main paper.
The authors promise to change this in the camera-ready version, and I trust them to a certain degree.
However, the extent of the changes is big enough to keep me from giving a clear acceptance without being able to read the revision.

**Limitations:**

Yes.

**Paper Formatting Concerns:**

No major formatting issues are noticed.

**Quality:**

2

**Strengths And Weaknesses:**

## Overall

In summary, the paper shows significant advancements in uncertainty quantification for large language models, an important research area, but it also comes short in the clarity of the presentation. Even though the core contributions seem to be strong enough, a plethora of minor issues make the work appear unfinished. Especially the almost non-existent discussion of TSU in the main part, which is sold as a contribution and used to motivate the proposed algorithm, seems to be problematic. In its current state, I cannot recommend this work to the NeurIPS audience, but I am looking forward to an improved version.

## Strengths
- Flexible and novel probabilistic framework applicable to natural language processing
- Important and relevant application
- Simple algorithm
- Extensive evaluation and comparison of baselines and datasets
- Very strong empirical performance

## Weaknesses
- Temperature Sensitivity of Uncertainty (TSU) is highlighted as a contribution and takes up a large part of the evaluations, but is never clearly motivated and explained beyond a few sentences in the main paper. The authors should decide whether they want to properly include TSU in the main part (which comes with a proper explanation) or if its results are moved to the appendix, where they have enough space for the explanation.
- There seems to be confusion about what calibration-based metrics are. A calibration-based metric quantifies in some way how close predicted probabilities are to ground-truth probabilities (c.f. [1]). The Brier score partly quantifies this; however, the AUROC and PRR do not. These are rejection-based metrics, which assess how informative an uncertainty metric is with respect to a correctness measure. Based on my understanding, calibration is not part of this research direction and should not be mentioned in this work (without a very extensive discussion).
- Discussion of theoretical foundations misses references. There is a large corpus of literature on random walks, semantic similarities, and tangent spaces; however, almost nothing is cited here. The theoretical framework is labelled as a major contribution, but it is not possible to put it into relation with other theoretical frameworks without actively searching for related theory outside of this work.
- Some of the algorithmic decisions seem ad hoc without theoretical connections, e.g., it is not made clear what theoretical quantities Monte Carlo and bootstrapping aim to estimate; why use bootstrap sampling with S samples when we generate r samples from the model? Why not use the r samples directly?
- I am not sure including evaluations with WD-px-py and MAX-py in the main paper makes sense, since they are not properly discussed, and considering there are more fundamental discussions (like TSU) missing

### Minor issues
- Line 30: It would be nice to spell out UQ the first time it is mentioned (besides the abstract)
- Line 52-65: Contributions could be more concise and compressed
- Line 74-79: Missing references for input perturbations and equivalence classes
- Line 82: ';' is an atypical set notation (use ':' or '|')
- Line 80-87: Missing reference for tangent space and missing justification for gaussian distribution
- Eq 3: Add bracket for matrix/vector multiplication to highlight order of operators when indexing (you did it right in Eq 4)
- Line 160: that -> there
- Line 169-172: This paragraph is very important. Maybe emphasize more? Further, the notation seems strange: First, we are not defining; we are making statements (so no delta). Second, it seems inconsistent to write $\mathbb{P} ( Y = y_j \mid X = x_i ) = \mathbb{P} ( y_j \mid x_i )$. Either $x_i$ and $y_j$ are constants or random variables, but using them as both seems confusing.
- Eq 5: Flip the side of the $=$, since $H(Y|X) := - \sum_i P(Y=y_i|X=x_i) \log P(Y=y_i|X=x_i)$ is the known definition (and provide citation)
- Line 176: Citation for wasserstein distance would be nice since it is used later on
- Eq 6: $S$ is already used as variable (although with a subscript). A different variable would be nice
- Algorithm 1: The following inputs (as far as I have understood) are missing: r, S, $\delta$, $\tau$, T
- Fig. 3: text in fig too small. Maybe the fig margins are also too small?
- Line 233: There are already plenty of baselines in use, which is immaculate; however, a work [2] from last year's ICML clearly outperformed semantic entropy and P(True) on almost similar benchmarks. Considering that KLE is more recent and used as a baseline, the method of [2] should be at least cited.
- Line 239-240: Please correct "P(Y |X) derived in Sec. 2.3 instead of P(Y |X)"
- Line 241: Add an explanation of why comparing the marginal distributions of input and output makes sense
- Line 243: AUROC and PRR are not calibration metrics
- Line 248: "map map"
- Line 268: The Brier score allows to compare models, so we can say that model A is likely better calibrated than model B (even though the Brier score is not only about calibration but also about sharpness, c.f. [3]). But we cannot say under what threshold of the Brier score a model is well-calibrated. Please reformulate.
- Table 1: Please explain how temperature 0.7 was chosen
- Table 2: Too bloated for main part, might fit better in appendix
- Figure 4: The figure presents important insights, but the font is too small to read


**References**

[1] Vaicenavicius, J., Widmann, D., Andersson, C., Lindsten, F., Roll, J., & Schön, T. (2019). Evaluating model calibration in classification. In The 22nd international conference on artificial intelligence and statistics (pp. 3459-3467). PMLR.

[2] Gruber, S. G., & Buettner, F. (2024). A Bias-Variance-Covariance Decomposition of Kernel Scores for Generative Models. In International Conference on Machine Learning (pp. 16460-16501). PMLR.

[3] Bröcker, J. (2009). Reliability, sufficiency, and the decomposition of proper scores. Quarterly Journal of the Royal Meteorological Society: A journal of the atmospheric sciences, applied meteorology and physical oceanography, 135(643), 1512-1519.

---

> ### Author Rebuttal · Authors · 2025-07-30
>
> First, we would like to thank you for your detailed and thoughtful review. We appreciate your recognition of the significance and originality of our work, as well as your positive assessment of our empirical results and the flexibility of our proposed framework.
>
> >**W1: TSU**
>
> Your suggestion regarding the placement of the TSU definition is very well received. Due to space constraints and our decision to prioritize the modeling framework, we initially placed TSU in the appendix rather than the main paper. Its simplicity and generality are precisely why we introduced it. We will bring the definition and motivation into the beginning of section 3 of the main paper as you recommend.
>
> >**W2: Calibration-based Metrics**
>
> Thank you for pointing out the naming issue. In the final version, we will revise all mentions of “calibration-based metrics” to the more appropriate term “correctness-based metrics” when referring collectively to AUROC, PRR, and Brier score. We include the Brier score not only because it partially reflects calibration ability, but more importantly because it enables a more fine-grained assessment of the alignment between predicted confidence (the negative of the uncertainty measure) and correctness labels. While AUROC and PRR are ranking-based metrics, Brier also accounts for the magnitude of the predicted confidence, offering a richer evaluation of how well uncertainty scores correspond to actual correctness.
>
> [Minor Issues-line 268] Regarding our statement in line 268, we will revise the sentence to:
>
> “Inv‑Entropy attains the lowest Brier score, indicating better calibration than the other baselines.”
>
> >**W3: Theoretical references**
>
> We add the following discussion in Section 2.1:
>
> "Similarity-based graphs are widely used in structure discovery [6] and feature alignment [7]. Recent work [4,5] leverages graph statistics to quantify LLM uncertainty using only model outputs. Our proposed Inv-Entropy further introduces perturbations in the input space and evaluates LLM uncertainty by assessing the alignment between the input similarity graph and the output similarity graph.
>
> The use of random-walk-based metrics for graph alignment was first introduced in [8], and we are, to the best of our knowledge, the first to apply such techniques to UQ in LLMs."
>
> [6] "Laplacian eigenmaps for dimensionality reduction and data representation." Neural computation, 2003).
>
> [7]  "Latent common manifold learning with alternating diffusion: analysis and applications." Applied and Computational Harmonic Analysis, 2019.
>
> [8] "Graph kernels." The Journal of Machine Learning Research, 2010.
>
> >**W4: Bootstraping and Monte-Carlo**
>
> We clarify that bootstrapping and Monte Carlo estimation are not ad hoc, but are both theoretically motivated components designed to estimate a core quantity in our framework that integrates perturbations and replications:
>
> $$\mathbb{E}_{Y_n \sim \text{Replications}} [H(X_n \mid Y_n)]$$
>
> **Why not use all $r$ samples directly?**
> In our dual random walk formulation, each Markov chain operates over a state space of paired inputs and outputs, $(x_i, y_i)$, with the critical assumption that these pairs are in one-to-one correspondence. However, LLM output $y_i$ is stochastic and such randomness may significantly affect the estimate of $\text{P}_y$, which is a nonlinear function of $Y_n$. Therefore, we use $S$ replicates of LLM outputs to stabilize Inv-Entropy.
>
> Empirically, we explored an alternative approach during development, where all $r$ outputs are used simultaneously. This is called "NR-Inv-Entropy" in our Table 1 and 2. This method consistently underperforms our proposed bootstrapping strategy, Inv-Entropy, across all evaluation metrics, confirming the concerns about output randomness and estimate stability.
>
> **Why use Bootstrapping with Monte Carlo averaging?**
> For each bootstrapped set indexed by $s$, we sample one output $y_i^{(s)}$ from the $r$ replications of each $x_i$, forming a valid output set $Y_n^{(s)} = \lbrace y_0^{(s)}, \dots, y_n^{(s)} \rbrace$.
>
> However, since each $Y_n^{(s)}$ reflects only one possible realization of outputs, its associated conditional entropy $H(X_n \mid Y_n^{(s)})$ captures uncertainty only for that particular bootstrapped set  $Y_n^{(s)}$, which is highly stochastic due to the inherent randomness of LLM. To comprehensively estimate the expectation over all possible LLM output realizations, we apply Monte Carlo averaging:
>
> $$\widehat{H}(X \mid Y) = \frac{1}{S} \sum_{s=1}^{S} H(X_n \mid Y_n^{(s)}).$$
>
> We find this Monte Carlo estimator to be numerically stable than "NR-Inv-Entropy".
>
>
> >**W5: Including WD-px-py and MAX-py**
>
> We included WD-px-py and MAX-py to highlight the flexibility of our probabilistic framework, which naturally supports alternative uncertainty metrics beyond Inv-Entropy; such as those based on comparing input and output marginals. Both variants perform competitively, outperforming most standard baselines, which supports their inclusion as valuable design alternatives.
>
> To also address your later question: comparing the marginal distributions of inputs and outputs is meaningful in our context as they offer alternative approaches to capture the alignment between the diversity of semantic inputs and the spread of model outputs. If semantically distinct inputs lead to highly similar outputs, this mismatch can signal reduced epistemic uncertainty. Metrics like WD-px-py can capture this behavior within our dual random walk construction.
>
> We will add more detailed explanations in the revised manuscript.
>
> >**Q1 & Q2: Hyperparameter Selection & Holdout Set Usage**
>
> We emphasize that we did not perform any tuning on the test sets.
>
> All hyperparameters except for temperature were chosen using a held-out set of 100 samples from the TriviaQA dataset. These samples were excluded from all subsequent evaluations and are not included in any of the reported results. All experimental results in the paper, across all datasets, are computed using these fixed hyperparameters, ensuring that evaluations are conducted on proper hold-out sets. We also note that when perturbing using ChatGPT, the GAAP-specific hyperparameters ($\delta$, $\tau$, $T$) are not used.
>
> [Minor Issues-Table 1] The temperature was set to 0.7 because it is the default configuration for many publicly available LLMs. We adopted this setting to reflect realistic usage and ensure that our evaluation aligns with typical deployment scenarios.
>
> Additionally, we refer to our response to **Reviewer tjtF** (under "W2 & W4 Sensitivity Analysis"), where we conducted an ablation study showing that Inv‑Entropy remains robust across different hyperparameter settings.
>
> >**Q3: Lemma 2.1 Relationship with the Algorithm**
>
> Section 2.1 introduces a simplified theoretical prototype to justify the perturb-then-estimate foundation of Inv-Entropy. Lemma 2.1 shows that the variance of inputs mapping to a fixed output under a function $f$ can illustrate how sensitivity to input perturbations reflects predictive inexactness.
>
> The tangent plane is a mathematical abstraction of the class of semantic invariant inputs to LLMs. In practice, we perturb the input question to different mutants which should have similar responses. Vaguely speaking, Inv-Entropy operationalizes the notion of "tangent space" of LLM by perturbing prompts. This is the nature of "input semantic invariance". However, generating diverse yet exactly semantic equivalent input perturbations is almost impossible. That is why Inv-Entropy also introduces an input similarity graph and uses a random walk on such graphs to characterize small input semantic variations.
>
> We will add a detailed correspondence between Section 2.1 and realistic LLMs in the revised manuscript.
>
> >**Minor Issues**
>
> We appreciate the detailed list of minor issues and will address all of them in the final version, including clarifying notation, adding missing references, fixing typos, improving figures, and streamlining contributions and tables as suggested.
>
> >>**Line 233: Adding Kernel Entropy Baseline in Reference [2]**
>
> Thank you for pointing out the work in [2]. To fully address your concern, we added Kernel Entropy in [2] with 5 and 50 replications per input. The results are shown below with the same experiment setting as that of Table 1 in our main paper:
>
> | **Metric (↑/↓)**   | **Method**                  | **TriviaQA** | **SciQ**    |
> |--------------------|-----------------------------|--------------|-------------|
> | **AUROC (↑)**      | Kernel Entropy (5)  | 0.687        | 0.685       |
> |                    | Kernel Entropy (50) | 0.797        | 0.706       |
> |                    | Inv-Entropy (Ours)          | **0.810**    | **0.771**   |
> | **PRR (↑)**        | Kernel Entropy (5)  | 0.794        | 0.812       |
> |                    | Kernel Entropy (50) | 0.884        | 0.837       |
> |                    | Inv-Entropy (Ours)          | **0.920**    | **0.843**   |
> | **Brier (↓)**      | Kernel Entropy (5)  | 0.160        | 0.153       |
> |                    | Kernel Entropy (50) | 0.124        | **0.131**   |
> |                    | Inv-Entropy (Ours)          | **0.117**    | 0.149       |
> | **TSU (0.7, 1.4)** | Kernel Entropy (5)  | 55.56%       | 62.89%      |
> |                    | Kernel Entropy (50) | 84.62%       | 76.73%      |
> |                    | Inv-Entropy (Ours)          | **88.78%**   | **80.61%**  |
>
> We observe that while 50 replications improve performance compared to 5, our Inv-Entropy approach still outperforms across all metrics. In our updated version, we will cite [2].

---

> > ### Comment · Reviewer_pA2S · 2025-08-04
> >
> > Thank you for your extensive response. Indeed, my concerns have been addressed, and I am willing to adjust my score slightly according to that.
> > The issues regarding hyperparameters, literature, and calibration have been fully resolved in my eyes.
> > However, my major concern was regarding the clarity of the presentation (especially motivation and explanation of TSU), and I would really need to be able to read the revised manuscript to feel comfortable in recommending acceptance.
> > Since the NeurIPS rebuttal phase does not support this, and since I am not the only reviewer who had major issues with the clarity, I cannot increase my score further. I am very sorry for this, and I am sure the revised work has a significantly higher chance of acceptance upon resubmission in another conference.

---

> ### Author Response · Authors · 2025-08-04
> **Follow-up**
>
> Dear Reviewer pA2S,
>
> We are very happy to hear that the issues regarding hyperparameters, literature, and calibration have been fully resolved from your perspective. Thank you for your thoughtful engagement!
>
> >**Regarding the revised manuscript**
>
> As you know, we have no control over the NeurIPS policy that disallows uploading a revised version during the rebuttal phase. We sincerely hope that this policy does not negatively affect your assessment of our work. As you know, the camera-ready version is intended to address such concerns, and we will move the details about TSU from the appendix to the main paper to fully address your concern.
>
> To help address your concerns, we have included below (at the end of this comment) the wording we intend to add to the main text regarding TSU in the revised manuscript.
>
> We would also like to kindly note that TSU is not the central contribution of the paper. This, in addition to its simplicity, is why we placed it in the appendix. Our central contribution lies in the proposed probabilistic framework and its state-of-the-art performance across multiple metrics and in comparison to various benchmarks.
>
> >**Regarding clarity**
>
> With respect to clarity and broader reviewer feedback, we would like to respectfully highlight that several reviewers explicitly praised the presentation, theoretical contributions, and clarity of our paper:
>
> - **Reviewer Q6Vh** called the paper *“well written and easy to follow”*, and emphasized the *“highly valuable theoretical perspective.”*
>
> - **Reviewer uG7q** described the paper as *“exceptionally clearly written”* and *“elegant,”* awarding it top marks for both quality and originality.
>
> - **Reviewer tjtF** stated that the paper was *“well-written and easy to follow,”* noted that the modeling section was *“clear and understandable,”* and praised both the motivation and empirical results.
>
> Reviewer cePH was the only reviewer who expressed concerns about clarity. We believe these concerns may stem from field-specific terminology rather than a lack of rigor or explanation. For instance, the reviewer questioned terms such as “semantic input” and “semantic embedding”, which are standard in the representation learning and NLP literature. We fully understand, however, that these terms may not be immediately accessible to readers from other methodological communities.
>
> We are very grateful for your engagement and sincerely hope our clarifications help strengthen your confidence in the revised work.
>
> >**TSU discussion to be brought from appendix to main paper**
>
> Using correctness as a proxy for uncertainty quantification (UQ) has clear limitations in open-ended or weakly supervised settings where correctness is unavailable or ill-defined. To circumvent this, we introduce a new evaluation metric: **Temperature Sensitivity of Uncertainty (TSU)**, which captures the proportion of instances where uncertainty increases consistently with temperature.
>
> This metric is motivated by the role of temperature in controlling the randomness of sampling in LLMs. Specifically, higher temperatures flatten the token distribution, increasing entropy and randomness, while lower temperatures make generation more deterministic [*Hinton et al. 2015*].
>
> Formally, let $\mathbb{T}_1 < \mathbb{T}_2 < \dots < \mathbb{T}_n$ be a sequence of temperature values. Then TSU is defined as:
>
>
> Formally, TSU is defined as:
> $TSU(\mathbb{T}_1, \mathbb{T}_2, \dots, \mathbb{T}_n) = \frac{1}{|\mathcal{D}|} \sum \mathbb{I}\left( \text{UQ}(S_x, \mathbb{T}_1) < \text{UQ}(S_x, \mathbb{T}_2) < \cdots < \text{UQ}(S_x, \mathbb{T}_n) \right)$
>
>
> where $\mathcal{D}$ denotes the dataset, $S_x$ represents a question or input in the dataset, $\text{UQ}(S_x, \mathbb{T})$ is the uncertainty score (e.g., Inv-Entropy) evaluated at temperature $\mathbb{T}$, and $\mathbb{I}(\cdot)$ denotes the indicator function.
>
> Importantly, TSU is agnostic to the output label $y$ and depends only on the input $S_x$, making it applicable even when "ground truth" is undefined or unavailable.
>
> Moreover, TSU goes beyond binary correctness evaluation by assessing how finely a method can track the gradation of uncertainty in response to controlled perturbations in temperature. This provides a richer and more generalizable criterion for UQ methods.

---

> > ### Comment · Reviewer_pA2S · 2025-08-05
> >
> > Thanks for pointing out the positive feedback from other reviewers.
> > I was aware of that, and I honestly disagree based on the submitted version.
> >
> > My main concern regarding clarity was that it would take substantial additional space to improve the camera-ready version. I now remember that you will get an additional page.
> > Given this additional page, I trust your claim that you will thoroughly explain TSU and insert connections with background literature that your methodology builds upon.
> > I like your contributions, and the evaluations seem thorough (it would be nice to add the kernel entropy results at least to the appendix).
> > I now tend towards acceptance, but I really cannot give a clear acceptance without reading a revised manuscript, which is simply an inherent limitation of the NeurIPS rebuttal system.

---

> > > ### Author Response · Authors · 2025-08-05
> > >
> > > Dear Reviewer pA2S,
> > >
> > > Thank you again for your thoughtful engagement throughout the process. We are very encouraged to hear that you now tend toward acceptance.
> > >
> > > We completely understand and respect your position. We do not want to push beyond the rating you feel comfortable giving. That said, we want to emphasize that we will include the Kernel Entropy results in the main paper and will make every effort to ensure the final version fully addresses your suggestions, including moving the TSU discussion into the main text.
> > >
> > > We genuinely wish we could have shown you the revised version during rebuttal, but unfortunately NeurIPS policy prevents that.
> > >
> > > Warm regards,
> > > The Authors
> > >
> > >
> > > Warm regards,

---

### Official Review · Reviewer_tjtF · 2025-06-27

**Clarity:** 3
**Significance:** 3
**Originality:** 3
**Rating:** 4
**Confidence:** 4

**Summary:**

Although there are lots of uncertainty quantification methods which trying to measure the uncertainty of Large Language Model, they are usually heuristic. To address this issue, in this paper, they propose a new uncertainty quantification framework called Inv-Entropy which is based on the perturbation algorithm called GAAP. Before suggesting the method, the authors raise motivations and Lemma, utilizing a simple 1d case which helps readers to understand more clearly.

**Questions:**

See the Weaknesses section.

**Ethical Concerns:**

["NO or VERY MINOR ethics concerns only"]

**Final Justification:**

I am satisfied with the author's response.

**Limitations:**

See the Weaknesses section.

Overall I like the paper and I am willing to increase my rating after seeing the response from the authors.

**Paper Formatting Concerns:**

Nothing

**Quality:**

3

**Strengths And Weaknesses:**

Strengths
- The paper is well-written and easy to follow
- The motivation is clear and give appropriate justification of their proposed method.
- Description of their methods, especially for the modeling part of their conditional probability and multiple sampling parts, are clear and understandable.
- Experimental results show that Inv-Entropy clearly outperforms other methods.

Weaknesses
- The most clear and straightforward concern about the proposed method is the computational cost. It would be helpful to include an analysis of the exact computational time and memory cost.
- This method seems to rely quite heavily on certain hyperparameters, such as $n$ and $S$. It would be helpful to see how robust the method is through an ablation study on these parameters.
- It would be helpful for understanding if there were practical entropy analyses or qualitative examples showing cases where the model outputs different $y$ values for similar $x$, or similar $x$ values for different $y$.
- Including an ablation study on the augmentation used in the GAAP method would help readers better understand the appropriate level of augmentation needed to estimate $\hat{H}$. It would also be beneficial to add ablation studies on the GAAP method's hyperparameters, $\delta$ and $\tau$, to analyze their impact. Additionally, incorporating ablations on cases without crossover or without mutation would further aid in understanding the individual contributions of these components.

---

> ### Author Rebuttal · Authors · 2025-07-30
>
> First, we would like to thank you for your constructive review. We appreciate your positive remarks regarding the clarity of our writing, the strength of our motivation and theoretical foundation, and the performance of our proposed Inv-Entropy method. We also sincerely thank you for noting that you like the paper and are open to increasing your rating.
>
> Below we provide an itemized response to your suggestions & concerns:
>
> >**W1: Computational Needs**
>
> Great suggestion. We would like to point you to our response (under "W2: Computational Needs") to **Reviewer Q6Vh**, who raised a similar question, where we also provide supporting experimental results. Overall, our method is highly efficient in the post-processing stage, after obtaining responses from the perturbations via multiple API calls. The main computational cost arises from the API calls themselves; however, responses for perturbed inputs can be generated in parallel, making the process significantly more scalable in practice.
>
> >**W2: Sensitivity Analysis on Hyperparameters**
>
> Thank you for the insightful suggestion. To fully address your comment, we have now run ablation studies on the parameters you highlighted.
>
> >>**The number of perturbations n**:
>
> | Metric         | n = 3 | n = 6 | n = 9 |
> |----------------|-------|-------|--------|
> | AUROC          |   0.725 ± 0.063    |   0.815 ± 0.052    |  0.837 ± 0.047      |
> | PRR            |   0.813 ± 0.057    |   0.884 ± 0.049    |    0.904 ± 0.042    |
> | Brier          |    0.145 ± 0.022   |   0.122 ± 0.021    |     0.115 ± 0.020   |
>
> The table above shows the effect of perturbation times $n$ on Inv-Entropy using ChatGPT-3.5-Turbo on TriviaQA with $r=5$. We observe that as the number of perturbations $n$ increases, the performance of Inv-Entropy improves consistently across all metrics. Notably, even with as few as 3 perturbations, the method still achieves competitive results, indicating robustness to the choice of $n$.
>
> >>**The number of replications r**:
>
> | **Metric** | **TriviaQA**        |                   |                   | **SciQ**           |                   |                   |
> |------------|---------------------|-------------------|-------------------|--------------------|-------------------|-------------------|
> |            | *r = 2*           | *r = 4*           | *r = 6*           | *r = 2*            | *r = 4*           | *r = 6*           |
> | **AUROC**  | 0.812        | 0.815      | 0.820      | 0.794       | 0.801      | 0.806      |
> | **PRR**    | 0.916       | 0.915      | 0.915      | 0.888      | 0.888      | 0.892     |
> | **Brier**  | 0.128      | 0.121      | 0.123     | 0.143      | 0.140     | 0.136      |
>
> This table presents the performance of Inv-Entropy on the TriviaQA and SciQ datasets using ChatGPT-3.5-Turbo as the number of replications $r$ increases. Although performance improves with larger $r$, the gains are relatively small. The results with $r = 2$ are already strong, confirming our claim in the "Computational Needs" that our framework does not rely heavily on a large number of replications. Compared to the ablation study on the number of perturbations $n$, we find that model performance is much more sensitive to changes in $n$ than in $r$.
>
> **These results also shed light on our theoretical findings, which highlight the importance of perturbation for uncertainty quantification**.
>
> >>**The temperature t**:
>
> | **Metric** | **TriviaQA**     |             |             | **SciQ**         |             |             |
> |------------|------------------------|-------------------|-------------------|------------------------|-------------------|-------------------|
> |            | *t = 0.3*        | *t = 1.0*   | *t = 1.4*   | *t = 0.3*        | *t = 1.0*   | *t = 1.4*   |
> | **AUROC**  | 0.870            | 0.827       | 0.810       | 0.767            | 0.754       | 0.795       |
> | **PRR**    | 0.939            | 0.936       | 0.903       | 0.900            | 0.864       | 0.870       |
> | **Brier**  | 0.085            | 0.117       | 0.132       | 0.151            | 0.153       | 0.146       |
>
> The table above presents an ablation study on the effect of temperature $t$ in the response sampling process for Inv-Entropy, evaluated on the TriviaQA and SciQ datasets using ChatGPT-3.5-Turbo. We observe that lower temperatures lead to better performance across all metrics. This suggests that reducing randomness in the model’s outputs helps stabilize the uncertainty estimation and improve reliability. Nonetheless, Inv-Entropy remains effective across a range of temperature settings, demonstrating its robustness.
>
> >>**The bootstrapping iterations S**:
>
> Please refer to Table 6 in the appendix. We observe that increasing S initially enhances the performance of our method. However, beyond a certain threshold $S=10$, further increases yield diminishing or no significant returns. Therefore,
> using more than 10 iterations appears to provide limited additional benefit.
>
>
> >**W3: Qualitative examples for responses with similar x and different y**
>
> The following table presents a representative case from the TriviaQA dataset, where semantically similar questions yield highly diverse answers.
>
> |          | original question                                                  | perturbed question1                                                     | perturbed question2                                                               | perturbed question 3                                                      |
> |--------------------|--------------------------------------------------------------------|---------------------------------------------------------------------------|--------------------------------------------------------------------------------------|------------------------------------------------------------------------------|
> |          | In 1999 Anna Kournikova signed a lucrative contract to model what? | What did Anna Kournikova sign a profitable agreement to showcase in 1999? | In 1999, Anna Kournikova agreed to a well-paying contract to promote which product? | Anna Kournikova inked a financially rewarding deal in 1999 to endorse what? |
> | Answer 1 | Sports Illustrated Swimsuit Issue                                  | Omega Watches                                                             | Taco Bell                                                                           | Adidas                                                                      |
> | Answer 2 | Sports Illustrated Swimsuit Issue                                  | Adidas                                                                    | Berlei Bras                                                                         | Adidas                                                                      |
> | Answer 3 | Sports Illustrated Swimsuit Issue                                  | K-Swiss                                                                   | Tennis shoes                                                                        | Adidas                                                                      |
> | Answer 4 | Sports Illustrated Swimsuit Issue                                  | Adidas                                                                    | Tennis shoes                                                                        | Adidas                                                                      |
> | Answer 5 | Sports Illustrated Swimsuit Edition                                | Omega Watches                                                             | K-Swiss                                                                             | Adidas                                                                      |
>
> Answers 1 through 5 correspond to independent replications for each question. This table clearly shows the value of our **perturbation-based approach**. If one only relies on replications and compares multiple outputs directly, as in other methods, the underlying uncertainty in this case would not be captured. By introducing semantically meaningful perturbations, our method is able to reveal this uncertainty and provide a more informative uncertainty estimate.
>
>
>
> >**W4: Sensitivity analysis on GAAP**
>
> | Metric   | $\delta = 0.3$ | $\delta = 0.5$ | $\delta = 0.7$ | $\delta = 0.9$ |
> |----------|----------------|----------------|----------------|----------------|
> | AUROC    | 0.723          | 0.752          | **0.843**      | 0.740          |
> | PRR      | 0.792          | 0.843          | **0.917**      | 0.801          |
> | Brier    | 0.165          | 0.143          | **0.103**      | 0.161          |
>
> This table reports the performance on the TriviaQA dataset using ChatGPT-3.5-turbo. As the similarity threshold $\delta$ decreases, more loosely related perturbated questions are included. Among the settings, $\delta = 0.7$ achieves the best overall performance, striking a balance between semantic diversity and fidelity to the original question. In contrast, overly high or low thresholds result in worse performance.
>
>
> **Regarding the comment on isolating the effects of crossover and mutation**, we note that our GAAP algorithm integrates these operations in a tightly coupled manner, making it non-trivial to disable one without disrupting the overall perturbation quality or validity. We will clarify this more explicitly in the final version.

---

> > ### Comment · Reviewer_tjtF · 2025-08-01
> >
> > Thank the authors for the detailed explanations. I'll finalize my score after watching all the discussions with authors and reviewers.

---

> ### Author Response · Authors · 2025-08-05
> **Follow-up**
>
> Dear Reviewer tjtF,
>
> Thank you for your engagement. We sincerely appreciate your positive assessment of our work.
>
> We hope our rebuttal addressed your suggestions and concerns, and we would be happy to provide any additional clarification or simulation results if needed.
>
> Warm regards
>
> The Authors

---

> > ### Comment · Reviewer_tjtF · 2025-08-08
> >
> > Thank authors for the engagement in the discussion, I appreciate all the detailed explanations and the discussion with me and other reviewers. Although there still exists some negative opinion, I want to keep my score positive.

---

> > > ### Author Response · Authors · 2025-08-08
> > > **Thank you**
> > >
> > > Dear Reviewer tjtF,
> > >
> > > We are glad our responses have helped address your suggestions and concerns, and we remain happy to provide any additional clarification or simulation results if that would be useful.
> > >
> > > Thank you again for your thoughtful engagement. The suggestions you provided have helped us strengthen the paper, and we truly appreciate that.
> > >
> > > Warm regards,
> > >
> > > The Authors

---

### Official Review · Reviewer_uG7q · 2025-06-29

**Clarity:** 4
**Significance:** 3
**Originality:** 4
**Rating:** 5
**Confidence:** 5

**Summary:**

The authors propose a new framework for disentangling aleatoric and epistemic uncertainty in settings where one can perturb input prompts. The method effectively unifies semantic entropy and input perturbation-based approaches, permitting more sophisticated judgements about a model's uncertainty (capturing both cases where one particular input results in many semantically related outputs and also those where one particular set of outputs is yielded by many semantically distinct questions). The authors conduct extensive experiments using one instantiation of their framework and show that it outperforms common baselines at the task of predicting correctness.

**Questions:**

Why is the TriviaQA AUROC for semantic entropy in Table 1 so low? In that paper they report TriviaQA AUROCs of > 0.8 (granted, using OPT).

How does the new method compare to a simple ensemble of existing semantic entropy and input perturbation based approaches?

**Ethical Concerns:**

["NO or VERY MINOR ethics concerns only"]

**Final Justification:**

I wouldn't say this paper reaches the level of "groundbreaking impact," but it's very strong and I hope it is accepted.

**Limitations:**

As I mentioned, the paper needs to discuss limitations in more detail.

**Quality:**

4

**Strengths And Weaknesses:**

Strengths:

The method is general, combines elements of several baselines from the literature (as I mention above), and is overall quite elegant. The paper is exceptionally clearly written and the experimental results also seem quite strong. I really like this paper.

Weaknesses:

Some small gripes:

1. I think both figures are quite opaque. Figure 1 depicts level sets as concentric, which is not really accurate. Figure 2 is also pretty confusing; it would help to add indices to the caption.
2. The paper should really include a comparison of the runtimes of different baselines. How much of the benefit of the new approach comes from simply spending more on compute?
3. Related work and limitations sections are both pretty sparse (yes, there's related work in the appendix, but it would be nice to make room for more in the main paper). An obvious limitation is that this is only tested (and only really applicable at all) on short-form Q/A.
4. Some of the evaluated tasks (e.g. TriviaQA) exhibit very little aleatoric uncertainty, and I imagine that similarity scores between different outputs on these tasks are systematically low. It would be nice to see this special case worked out theoretically.

---

> ### Author Rebuttal · Authors · 2025-07-29
>
> First, we would like to thank you for the encouraging and thoughtful review. We sincerely appreciate your positive feedback on the generality, clarity, and elegance of our framework, and we’re glad the experimental results resonated with you.
>
>
> >**W1: Figures 1 \& 2**
>
> Thank you for the helpful feedback. For Figure 1, we will revise the illustration to eliminate the misleading appearance of concentric level sets and clarify that it is meant as a conceptual sketch. For Figure 2, we will add a detailed explanation of the transitions involving the highlighted nodes in the final version, to help readers grasp our model framework more intuitively.
>
> >**W2: Computational Needs**
>
> Great suggestion. We would also like to point you to our response (under "W2: Computational Needs") to **Reviewer Q6Vh**, who raised a similar question, where we also provide supporting experimental results. Overall, our method is highly efficient in the post-processing stage, after obtaining responses from the perturbations via multiple API calls. The main computational cost arises from the API calls themselves; however, responses for perturbed inputs can be generated in parallel, making the process significantly more scalable in practice.
>
> >**W3: Related Work in the Appendix**
>
> As you suggested, we will bring the related work section of the appendix into the final version of the main paper.
>
> >**W3: Applicability to Long Form Q/A**
>
> You raise a great point. Indeed, our inverse perspective for defining the final metric was designed with short answers in mind; since the shorter the answer, the more important it becomes to explore the diversity of the input space. That said, our probabilistic framework is general and can be applied to any answer format, with various metrics derivable from it. For instance, in the paper, we also present results using non-inverse entropy as well as the Wasserstein distance between $p(x)$ and $p(y)$. Exploring how different metrics perform across answer lengths is a promising direction we leave for future work.
>
> To fully address your concern, we have added additional experiments on a long-form answer dataset, where we study the method’s sensitivity to answer length. **These results are provided in our rebuttal to Reviewer Q6Vh**. Our experiments show that the method remains effective in long-form QA settings. Due to space constraints, we refer you to our response to **Reviewer Q6Vh** (under "W1 & Q1: New experiments on a long-form answer dataset"), who raised a similar question.
>
> >**W4: Extreme-Case Analysis: Small or no Aleatoric Uncertainty**
>
> If we understand your question correctly, you are asking us to consider what happens in cases with very little aleatoric uncertainty; for instance, when perturbations of an input yield the same output each time. To address your concern, we attempt to understand this scenario mathematically by analyzing an extreme case in which perturbing an input always yields exactly the same output. This case may not be uncommon in tasks involving very short-form answers.
>
> Mathematically, suppose all outputs are identical, even after perturbing the inputs: $y_0 = y_1 = \cdots = y_n$. As a result, all pairwise output similarities are maximized: $a_{\text{similarity}}(y_i, y_j) = 1$ for all $i, j$.
>
> **1. Identical Inputs.** Suppose all perturbed inputs are identical: $x_0 = x_1 = \cdots = x_n$, and all outputs are also identical: $y_0 = y_1 = \cdots = y_n$. Then $a_{\text{similarity}}(x_i, x_j) = a_{\text{similarity}}(y_i, y_j) = 1$ for all $i, j$, and both $\text{P}_x$ and $\text{P}_y$ become uniform matrices with entries $1/(n+1)$. The conditional distribution becomes $P(X = x_i \mid Y = y_j) = 1/(n+1)$, and the Inv-Entropy is maximized: $H(X \mid Y) = \log(n+1)$.
>
> **2. Slightly Varying Inputs.** If inputs differ slightly, say $a_{\text{similarity}}(x_i, x_j) = 1 - \epsilon$ for small $\epsilon$, then $P_x$ becomes nearly uniform, with diagonal entries slightly higher. As a result, $P(X \mid Y)$ deviates mildly from uniformity, and the entropy decreases slightly: $H(X \mid Y) = \log(n+1) - \mathcal{O}(\epsilon^2)$.
>
> **3. More Diverse Inputs.** When inputs are semantically diverse, $a_{\text{similarity}}(x_i, x_j)$ spans a wide range. The matrix $\text{P}_x$ becomes structured, and $P(X \mid Y)$ becomes more peaked. Consequently, Inv-Entropy drops further, reflecting reduced uncertainty due to input-output specificity.
>
> These regimes confirm that Inv-Entropy is sensitive to input variability even when outputs remain constant; precisely as intended by the inverse modeling perspective.
>
> **Thank you for raising this question. We have not thought of showing this analysis before and it largely enriches our paper**.
>
> >**Q1: Difference in Reported AUROC for Semantic Entropy on TriviaQA**
>
> Thank you for this insightful question. There are several key differences between our setup and that of the prior Semantic Entropy paper [1] that explain the discrepancy in AUROC values:
>
> First, the prior work used OPT, a white-box model with access to token-level logits. In contrast, we use ChatGPT-3.5-Turbo, a black-box model. While both implementations of Semantic Entropy cluster outputs by semantic meaning, the method for estimating cluster probabilities differs. The white-box version uses log-probabilities to weight clusters more precisely. The black-box version, which lacks access to logits, estimates cluster probabilities using relative frequency. This approximation is inherently less precise and tends to result in lower AUROC scores.
>
> Second, we use ChatGPT to assess whether a generated answer is semantically equivalent to the gold reference. The original Semantic Entropy paper, however, used ROUGE-L scores, which rely on lexical overlap and tend to overestimate correctness for similar-looking outputs.
>
> Third, we use instruction-style prompting uniformly across all methods. The prior work adopted few-shot prompting via in-context learning, which is known to enhance both accuracy and calibration, especially for white-box models.
>
>
> >**Q2: Adding Semantic Entropy–Perturbation Ensemble Baselines**
>
> Following your suggestion, we investigate two ensemble strategies for combining **Semantic Entropy** with input perturbations to enhance uncertainty estimation. We adopt the notation and perturbation setup introduced in Section 2.5 of the main paper.
>
> Let $S_{x_0}$ denote the original question prompt, and let $\text{Per}(S_{x_0}) = \lbrace S_{x_0}, S_{x_1}, \dots, S_{x_n} \rbrace$ be the set of $n+1$ semantically perturbed versions (including the original). For each perturbed prompt $S_{x_i}$, we sample $r$ responses from the language model, yielding response sets $R_i = \lbrace y_{i,1}, \dots, y_{i,r}\rbrace$.
>
> We consider the following:
>
> * **Semantic Entropy.**
>   The original method computes entropy solely over the $r$ responses to the original prompt $S_{x_0}$. No perturbation is used.
>
> * **Semantic Entropy (Joint).**
>   This ensembled  Semantic Entropy computes entropy over all responses from the original and perturbed prompts $ \mathcal{R}_{\text{joint}} = R_0 \cup R_1 \cup \dots \cup R_n$.
>
> * **Semantic Entropy (Average).**
>   An entropy score is computed independently for each response set $R_i$, and the final score is the average over all $i$.
>
> The results are shown below with the same experiment setting as that of Table 1 in our main paper:
>
> | **Metric (↑/↓)**         | **Method**                     | **TriviaQA**          | **SciQ**              |
> |--------------------------|--------------------------------|------------------------|------------------------|
> | **AUROC (↑)**            | Semantic Entropy      | 0.698 ± 0.048          | 0.611 ± 0.040          |
> |                          | Semantic Entropy (Joint)       | 0.783 ± 0.048          | 0.709 ± 0.050          |
> |                          | Semantic Entropy (Average)     | 0.784 ± 0.046          | 0.698 ± 0.053          |
> |                          | Inv-Entropy (Ours)             | **0.810 ± 0.051**      | **0.771 ± 0.051**      |
> | **PRR (↑)**              | Semantic Entropy     | 0.813 ± 0.040          | 0.725 ± 0.047          |
> |                          | Semantic Entropy (Joint)       | 0.877 ± 0.039          | 0.820 ± 0.028          |
> |                          | Semantic Entropy (Average)     | 0.873 ± 0.039          | 0.805 ± 0.027          |
> |                          | Inv-Entropy (Ours)             | **0.920 ± 0.030**      | **0.843 ± 0.054**      |
> | **Brier (↓)**            | Semantic Entropy     | 0.154 ± 0.019          | 0.168 ± 0.023          |
> |                          | Semantic Entropy (Joint)       | 0.134 ± 0.023          | 0.154 ± 0.018          |
> |                          | Semantic Entropy (Average)     | 0.140 ± 0.021          | 0.158 ± 0.018          |
> |                          | Inv-Entropy (Ours)             | **0.117 ± 0.018**      | **0.149 ± 0.022**      |
> | **TSU (0.7, 1.4)**       | Semantic Entropy      | 17.35%                 | 25.51%                 |
> |                          | Semantic Entropy (Joint)       | 59.18%                 | 61.86%                 |
> |                          | Semantic Entropy (Average)     | 63.27%                 | 73.20%                 |
> |                          | Inv-Entropy (Ours)             | **88.78%**             | **80.61%**             |
>
> Adding perturbation-based ensemble variants of Semantic Entropy improves its performance. However, Inv-Entropy still consistently outperforms all variants across all metrics and both datasets, confirming the strength of our approach.
>
>
> >**References**
>
> [1] "Semantic Uncertainty: Linguistic Invariances for Uncertainty Estimation in Natural Language Generation." The Eleventh International Conference on Learning Representations, 2023.

---

> ### Author Response · Authors · 2025-08-05
> **Follow-up**
>
> Dear Reviewer uG7q,
>
>
> Thank you once again for your thoughtful and supportive review. We sincerely appreciate your positive assessment of our work.
>
> We hope our rebuttal addressed your suggestions and concerns, and we would be happy to provide any additional clarification or simulation results if needed.
>
>
> Warm regards
>
> The Authors

---

### Official Review · Reviewer_Q6Vh · 2025-07-03

**Clarity:** 3
**Significance:** 4
**Originality:** 4
**Rating:** 5
**Confidence:** 2

**Summary:**

This paper studies uncertainty quantification of LLMs through the lens of LLM input-output behavior as a dual random walk. This leads to quantifying uncertainty in terms of the diversity of the inputs which lead to a particular output, rather than the other way around. The proposed uncertainty metric, Inv-Entropy, is shown to empirically outperform existing approaches across a variety of benchmarks.

**Questions:**

* How does the method perform for questions which long-form answers? If this is not in-scope, that is fine, but it should be stated that this method works for short-form questions only.
* Does the perturbation function always maintain semantic equivalence of the questions? The example in the Appendix shows "golfer" being replaced by "driver" which works in the shown question, but does not universally work as a replacement.

**Ethical Concerns:**

["NO or VERY MINOR ethics concerns only"]

**Final Justification:**

During the discussion, the authors provided additional results on a long form answer dataset which make me more confident in my assessment. I am also satisfied by their response regarding computation time and details of the perturbation function. Overall, after the discussion, I am very positive about the paper's acceptance.

**Limitations:**

yes

**Quality:**

3

**Strengths And Weaknesses:**

Strengths:
* Many papers in UQ for LLMs use perturbation-based approaches, but this paper makes a theoretical argument for why such an approach is reasonable. Their method then capitalizes on the perturbation approach by using both input and output perturbations, and it's effectiveness is supported by extensive empirical results comparing with many relevant baselines.
* The paper is well written and easy to follow.
* The theoretical perspective is highly valuable to this problem which is usually approached with empirically motivated methods.
* The approach of using an inverse model of entropy appears highly effective in practice and provides space for further work on how the uncertainty of the two random walks in the input and output space are quantified.
* The paper proposes a new metric for evaluating UQ for LLMs called TSU which does not use correctness as a proxy for uncertainty. This is useful for settings without ground truth.

Weaknesses:
* The method is not evaluated in a long-form answer setting. One of the baselines is specifically designed for such a setting, so if it is assumed that the method is only applicable to short-answer questions, then this should be stated.
* The limitation of perturbations being expensive, and the method being overall computationally expensive is acknowledged, but I did not see any analysis of how this compares to the baselines.
* The design of the perturbation function is based on replacing synonyms and other simple text modifications which seems like it could possibly lead to accidentally changing the semantics of the question.

---

> ### Author Rebuttal · Authors · 2025-07-29
>
> Thank you for the thoughtful review and encouraging feedback. We appreciate your recognition of the inverse perspective, the theoretical grounding via Lemma 2.1, and the clarity of the paper. We are also glad you found TSU valuable for evaluating UQ without ground truth.
>
> Below we provide an itemized response to your suggestions \& concerns:
>
> >**W1 & Q1: New experiments on a long-form answer dataset**
>
> This is a great suggestion. Indeed, our method is generic by nature and can be applied to both long and short answers. To fully address your concern, we have conducted additional experiments using the Natural Questions dataset [1], which includes both short and long-form answers. We randomly selected 1,000 questions that contain only long-form answers. The reference answers in this subset range from 34 to 350 tokens in length.
>
> To analyze performance across different answer lengths, we further divided the dataset into three subsets based on the number of tokens in the reference answers:
>
> * **Short** (<80 tokens): 280 samples
> * **Medium** (80–120 tokens): 320 samples
> * **Long** (≥120 tokens): 400 samples
>
> We evaluated both baseline methods and our proposed method on each of these subsets as well as the full 1,000-sample set. The results are shown below with the same experiment setting as that of Table 1 in our main paper:
>
> | **Metric (↑/↓)** | **Method**            | **Short** | **Medium** | **Long** | **Full** |
> |------------------|------------------------|-----------|------------|----------|---------|
> | **AUROC (↑)**    | Semantic Entropy       | 0.509     | 0.461      | 0.584    | 0.521   |
> |                  | VU                     | 0.531     | 0.495      | 0.508    | 0.533   |
> |                  | P(True)                | 0.529     | 0.473      | 0.548    | 0.519   |
> |                  | LexSim                 | 0.624     | 0.438      | 0.555    | 0.518   |
> |                  | DegMat                 | 0.547     | **0.621†** | 0.484    | 0.551   |
> |                  | LUQ                    | **0.662†** | 0.508      | **0.612** | **0.627†** |
> |                  | KLE                    | 0.265     | 0.456      | 0.445    | 0.410   |
> |                  | Inv-Entropy (Ours)     | **0.794** | **0.634**  | **0.589†** | **0.661** |
> | **PRR (↑)**      | Semantic Entropy       | 0.420     | 0.584      | 0.507    | 0.505   |
> |                  | VU                     | 0.489     | 0.615      | 0.495    | 0.537   |
> |                  | P(True)                | 0.427     | 0.584      | 0.478    | 0.502   |
> |                  | LexSim                 | 0.628     | **0.684†** | 0.544    | 0.563   |
> |                  | DegMat                 | 0.543     | 0.682      | **0.548** | 0.549   |
> |                  | LUQ                    | **0.649†** | 0.655      | **0.523†** | **0.595†** |
> |                  | KLE                    | 0.354     | 0.649      | 0.444    | 0.449   |
> |                  | Inv-Entropy (Ours)     | **0.747** | **0.742**  | 0.508    | **0.614** |
> | **Brier (↓)**    | Semantic Entropy       | 0.227     | 0.230      | 0.221    | 0.242   |
> |                  | VU                     | 0.209     | 0.218      | 0.216    | 0.223   |
> |                  | P(True)                | 0.225     | 0.232      | 0.230    | 0.244   |
> |                  | LexSim                 | 0.169     | 0.199**    | 0.207    | 0.225   |
> |                  | DegMat                 | 0.187     | **0.181†** | 0.213    | 0.229   |
> |                  | LUQ                    | **0.164†** | 0.209      | **0.179** | **0.208†** |
> |                  | KLE                    | 0.228     | 0.203      | 0.225    | 0.244   |
> |                  | Inv-Entropy (Ours)     | **0.125** | **0.175**  | **0.193†** | **0.201** |
>
>  **Bold** indicates the best performer. † indicates the second-best.
>
> Our method achieves state-of-the-art performance on the full dataset, ranking first across all three metrics. We also observe some sensitivity to answer length: it shows a clear and substantial advantage on the short-answer subset, highlighting the effectiveness of the inverse-design mechanism when responses are more concise. On the medium-length subset, it continues to outperform all baselines, though with a smaller margin. On the long-answer subset, while not the top performer, our method remains competitive and yields reasonable results compared to strong baselines such as LUQ.
>
> Intuitively, these results align with expectations under the inverse perspective: the shorter the answer, the more important it becomes to explore the diversity of the input space. Nevertheless, our approach consistently ranks among the top two methods even in the long-form QA setting. We end by noting that our probabilistic framework is also generic by nature and can accommodate a forward perspective or alternative uncertainty metrics beyond entropy computed over $p(X|Y)$. Exploring such metrics across different answer lengths is a promising direction we leave for future work.
>
> >**W2: Computational Needs**
>
> For non-locally hosted LLMs, uncertainty quantification methods generally involve two stages: Obtaining responses from the model via API calls and computing the uncertainty scores based on those responses. This applies to both our method and all baselines; the only exception is Verbalized Uncertainty (VU), which directly returns a score from the API without requiring post-processing. As discussed in the paper, any method that relies on perturbations and/or replications incurs additional computational cost. However, a major advantage is that responses for perturbed inputs can be generated in parallel, making the process more scalable in practice.
>
> With this in mind, we divide the computational cost into two components:
> (**A**) Computation time after responses are collected, and
> (**B**) Total cost, which includes (i) along with API call time and perturbation generation.
>
> **Regarding API cost**:
>
> Our method introduces $n$ perturbations per question and generates $r$ response replications for each version, resulting in a total of  $(n + 1) × r$ API calls per input. Thus, the cost scales linearly with both the number of perturbations and replications. That said, while our method introduces perturbations, it requires far fewer replications than several existing baselines. For instance, Semantic Entropy relies heavily on large $r$ values (with $n = 0$); prior work recommends $r > 10$ for stable performance (see [2]). In contrast, we demonstrate that even without replication ($r = 1$), our method remains effective. This is evidenced by the strong performance of **NR-Inv-Entropy**, which uses perturbation alone and still consistently outperforms many baselines.
>
>
> **Regarding computation after responses are collected**:
>
> Below, we present the compute results, without parallelization. The reported numbers are averaged over all sampled questions in TriviaQA. As shown, **our method is highly efficient in the post-processing stage** when computing Inv-Entropy scores. Furthermore, in settings where response generation time is negligible, such as when LLMs are deployed locally and perturbations are processed in parallel, our total computation cost is often lower than that of existing baselines.
>
> | Method                | Peak GPU Memory (MB) | Time A (s) | Time B (s) |
> |----------------------|----------------------|----------------------|-----------------------|
> | Semantic Entropy      | 3575.31              |      13.985          |      20.183                 |
> | VU                    | 0                    | 0                    |       0.620                |
> | P(True)               | 0                    |    0.001             |       2.201                |
> | LexSim                | 0                    |      0.022           |         3.189               |
> | DegMat                | 1610.29              |      6.143          |              15.905         |
> | LUQ                   | 1580.21              |      3.821           |             4.432          |
> | KLE                   | 1608.52              |      1.323           |             6.725          |
> | Inv‑Entropy (Ours)    | 86.65                | 1.990                 |   10.323                    |
> | NR‑Inv‑Entropy (Ours) | 86.65                | 1.769           |        6.358               |
>
> >**W3 & Q2: Perturbation function \& maintain semantic equivalence**
>
> We would like to start by noting that our framework and the Inv-Entropy metric are compatible with any perturbation algorithm. Notably, Table 1 in the main paper demonstrates our strong performance even when using ChatGPT-based paraphrasing as the perturbation method. As a form of rewriting rather than arbitrary perturbation, ChatGPT-based paraphrasing can be reasonably assumed to preserve semantic equivalence.
>
> Regarding our proposed GAAP approach, while we cannot guarantee perfect semantic equivalence, we include a parameter $\delta$ that controls the maximum allowable deviation from the original question to preserve closeness to the input. Interestingly, it is precisely this small deviation that allows GAAP to explore a broader input space, contributing to its superior performance compared to ChatGPT-based paraphrasing (see Figure 4 in the main paper).
>
> Additionally, in more sensitive applications, one could incorporate an external filtering mechanism (e.g., an LLM) to remove GAAP-generated perturbations that deviate too far semantically. Such heuristics serve as a practical safeguard against rare but problematic cases. We will revise the discussion in the final version of the paper to clarify this aspect more thoroughly.
>
> >**References**
>
> [1] "Natural questions: a benchmark for question answering research." Transactions of the Association for Computational Linguistics, 2019.
>
> [2] "Semantic Uncertainty: Linguistic Invariances for Uncertainty Estimation in Natural Language Generation." The Eleventh International Conference on Learning Representations, 2023.

---

> > ### Comment · Reviewer_Q6Vh · 2025-08-07
> >
> > Thank you for the response and additional experiments. This has adequately addressed my concerns and I will keep my score.

---

> > > ### Author Response · Authors · 2025-08-08
> > > **Thank you**
> > >
> > > Dear Reviewer Q6Vh,
> > >
> > > We are glad our response addressed your suggestions and concerns, and we would be happy to provide any additional clarification or simulation results if needed.
> > >
> > > Again, we sincerely appreciate your positive assessment of our work.
> > >
> > > Warm regards
> > >
> > > The Authors

---

### Author Response · Authors · 2025-08-09
**Thank you**

Dear Reviewers,

We sincerely thank you for your constructive feedback. We highly value the suggestions, encouragement, and positive comments acknowledging the novelty and significance of our work. We are also glad to have addressed the concerns you raised, and we remain eager to clarify any further points if needed.

The final version of our paper will reflect all improvements and suggestions made during the process.

Warm regards,

The Authors

---

### Note · Authors · 2025-08-11

Dear AC and Reviewers,

We immensely thank you for your time and highly insightful feedback. Upon your suggestions, we have made the following additions:

1. **Additional comparisons and evaluations**

We conducted all the requested evaluations, including:

- Comparison of computational cost
- Evaluations on new baseline models: (1) kernel entropy, (2) ensemble of semantic entropy and perturbation, and (3) Inv-entropy defined on the stationary distribution of our dual random walk
- Evaluation on a new long-form answer dataset
- Ablation studies on most of the hyperparameters we used
- A qualitative example

Across all these evaluations, our method outperformed the new baselines, was robust to all hyperparameter settings, and showed computational efficiency.

2. **Methodology reinforcement**

We expanded the discussion on the new evaluation metric, TSU, and elaborated on the motivation for adopting bootstrapping and Monte Carlo methods. We also added a theoretical discussion on extreme-case analysis, examining situations where outputs are identical and perturbations are either identical or different, and highlighted the corresponding theoretical implications for Inv-Entropy.

**Summary of Reviewer Feedback**

- We are pleased that all the concerns raised by the reviewers have been addressed, and **no further comments were raised**.

- **Reviewers Q6Vh, uG7q, tjtF, and pA2S** provided **highly positive feedback** and acknowledged that their comments were addressed. **Reviewer pA2S**, who initially had some concerns, stated after our response: “I like your contributions” and “I now tend towards acceptance”, but noted, “I really cannot give a clear acceptance without reading a revised manuscript.”  We understand this, but as NeurIPS does not allow reviewers to see a revised version, we hope not to be penalized for it.

- **Reviewer cePH** was the only one with initially negative feedback. We highly respect their comments and addressed all questions in detail. They initially had some misunderstandings about our method, but after our explanation, their follow-up noted that “many of my confusions have been clarified” and thanked us for clarifying our contribution. We deeply value their feedback but respectfully reiterate that the paper does not contain the technical issues initially suggested.

*The final version of our paper will reflect these strengths and the improvements made during the review process*.

Thanks again,

The Authors

---

### Decision · Program_Chairs · 2025-09-17

**Decision:**

Accept (poster)

**Comment:**

This paper proposes a fully probabilistic framework for uncertainty quantification in LLMs based on an inverse entropy perspective. The reviews for this paper were largely positive, though a clear consensus was not reached, making this a borderline case. Three reviewers (Q6Vh, uG7q, tjtF) recommended acceptance, noting the submission's theoretical grounding, originality, and strong empirical evaluation. Initial concerns from these reviewers regarding the scope (long-form QA), computational cost, and hyperparameter sensitivity were were discussed and reviewers appeared satisfied with the authors' responses. However, reviewers pA2S and cePH raised concerns about the clarity of the presentation and the rigor of the mathematical framework. Reviewer cePH remained unconvinced post-rebuttal, maintaining a recommendation for rejection due to what they perceived as a lack of rigor and motivation. While I sympathize with the points made by Reviewer cePH regarding the clarity of the mathematical presentation, I found the authors' explanations in the rebuttal to be clear. The level of rigor presented in the manuscript appears suitable for the UQ in LLMs community, which straddles theoretical motivations and strong empirical validation. In my assessment, the overwhelmingly positive feedback on the paper's novelty and performance from the other reviewers outweighs this dissenting opinion. I (weakly) recommend acceptance.